# A New, Physics-Based Continuous-Time Reinforcement Learning Algorithm with Performance Guarantees

## Abstract

We introduce a new, physics-based continuous-time reinforcement learning (CT-RL) algorithm for control of affine nonlinear systems, an area that enables a plethora of well-motivated applications. Based on fundamental input/output control mechanisms, our approach uses reference command input (RCI) as probing noise in learning. With known physical dynamics of the environment, and by leveraging on the Kleinman algorithm structure, our RCI-based CT-RL algorithm not only provides theoretical guarantees such as learning convergence, solution optimality, and closed-loop stability, but also well-behaved dynamic system responses with data efficiency during learning. Our results are therefore an advance from the two currently available classes of approaches to CT-RL. The first school of adaptive dynamic programming (ADP) methods features elegant theoretical results stemming from adaptive and optimal control. Yet, they have not been shown effectively synthesizing meaningful controllers. The second school of fitted value iteration (FVI) methods, also the state-of-the-art (SOTA) deep RL (DRL) design, has shown impressive learning solutions, yet theoretical guarantees are still to be developed. We provide several evaluations to demonstrate that our RCI-based design leads to new, SOTA CT-RL results.

## 1 Introduction and Related Work

Continuous-time optimal control problems can be found in many important engineering and socioeconomic application domains such as aerospace (Stengel, 2022), waste water treatment (Yang et al., 2022), robotics (Craig, 2005), and economics (Caputo, 2005). Reinforcement learning (RL) emerged as a systematic method in the early 1980s (Barto et al., 1983; Sutton & Barto, 1998) to combat the curse of dimensionality. While discrete-time (DT) RL algorithms (Si et al., 2004; Lewis et al., 2012; Kiumarsi et al., 2018; Bertsekas, 2017; Liu et al., 2021) have demonstrated extensive theoretical guarantees and demonstrations in applications, CT-RL algorithms have seen fewer theoretical results and little applications successes. Current CT-RL results generally fall into two classes: adaptive dynamic programming (ADP), and actor-critic deep RL (DRL), each discussed below.

The first school of adaptive dynamic programming (**ADP**) learns through optimal and adaptive control frameworks, oftentimes treating network weights as part of the adaptation parameters. ADP approaches were developed largely within the scope of seminal works such as integral reinforcement learning (IRL) (Vrabie & Lewis, 2009), synchronous policy iteration (SPI) (Vamvoudakis & Lewis, 2010), robust ADP (RADP) (Jiang & Jiang, 2014), and continuous-time value iteration (CT-VI) (Bian & Jiang, 2022). Refer to a recent comprehensive and systematic review of up-to-date ADP methods (Wallace & Si, 2022) for how peripheral algorithms revolve around the four central works. As a result of ADP's theoretical frameworks in adaptive and optimal control, Lyapunov arguments are available to prove qualitative properties including weight convergence and closed-loop stability results. Yet, quantitative results are few, as the proposed algorithms have only been evaluated on simple systems with known optimal solutions. For additional performance and design insights of ADP approaches, please refer to Appendix J.

The second deep RL (**DRL**) approach is the most recent and perhaps most promising to date, in particular fitted value iteration (FVI) deep RL methods. These frameworks solve the Hamilton-

Jacobi-Bellman/Isaacs (HJB/HJI) equations directly through approximation by deep networks, large datasets, and extensive training. The concept of applying black-box function approximation to solve the HJB traces back to the seminal work of Doya (2000). Subsequently, Tassa & Erez (2007) proposed a foundational value-function approximation framework using least-squares regression techniques for its neural network training. Recently, the novel continuous FVI (cFVI) (Lutter et al., 2023a) and robust FVI (rFVI) (Lutter et al., 2022) algorithms empirically exhibit low variance and control performance far surpassing that of prevailing ADP methods, and stand for the state-of-the-art currently. However, theoretical guarantees as those offered by ADP are yet to be developed.

**Contributions.** We propose a new model-based learning approach with the following three contributions to CT-RL: 1) Our novel reference command input (RCI) learning framework leverages fundamental input/output control mechanisms for well-behaved dynamic system responses and learning performance guarantees. 2) We take advantage of Kleinman's structure and data-driven learning to address nonlinear learning control problems with data efficiency. 3) This model-based learning framework presents new SOTA results through our performance evaluations.

In addition, when the system physics afford a decentralized structure with distinct dynamical loops, RCI can use this information to reduce dimensionality and make the algorithm even more efficient.

## 2 METHOD

RCI addresses the **same** affine nonlinear system as the above SOTA ADP and DRL methods:

$$\dot{x} = f(x) + g(x)u, \tag{1}$$

where $x \in \mathbb{R}^n$ is the state, $u \in \mathbb{R}^m$ is the control, $f : \mathbb{R}^n \to \mathbb{R}^n$, and $g : \mathbb{R}^n \to \mathbb{R}^{n \times m}$. As standard, we assume $f$ and $g$ are Lipschitz on a compact set $\Omega \subset \mathbb{R}^n$ containing the origin $x = 0$ in its interior, and that $f(0) = 0$. We consider the infinite-horizon undiscounted cost

$$J(x_0) = \int_0^\infty (x^T Q x + u^T R u) \, d\tau, \tag{2}$$

where $Q \in \mathbb{R}^{n \times n}$, $Q = Q^T \geq 0$ and $R \in \mathbb{R}^{m \times m}$, $R = R^T > 0$ are the state and control penalties.

**Background: Kleinman's Algorithm for Linear Systems (Kleinman, 1968).** We adapt some successive approximation concepts from Kleinman's algorithm to the proposed nonlinear RCI algorithm for data efficiency. Classical Kleinman's algorithm considers the linear time-invariant system $\dot{x} = Ax + Bu$. We assume that $(A, B)$ is stabilizable and $(Q^{1/2}, A)$ is detectable (Rodriguez, 2004). Kleinman's algorithm iteratively solves for the optimal LQR control $K^* \in \mathbb{R}^{m \times n}$ as follows. For iteration $i = 0, 1, \ldots$, on the current policy $K_i$, let $P_i \in \mathbb{R}^{n \times n}$, $P_i = P_i^T > 0$ be the solution of the algebraic Lyapunov equation (ALE)

$$(A - BK_i)^T P_i + P_i(A - BK_i) + K_i^T R K_i + Q = 0. \tag{3}$$

Then, $P_i$ solved from (3) leads to the new policy $K_{i+1} \in \mathbb{R}^{m \times n}$ as

$$K_{i+1} = R^{-1} B^T P_i. \tag{4}$$

**Definition 2.1 (Operators for Learning)** For $P = P^T \in \mathbb{R}^{n \times n}$, define its "vectorization" $v(P)$ as

$$v(P) = \begin{bmatrix} p_{11}, & 2p_{12}, \ldots, & 2p_{1n}, & p_{22}, & 2p_{23}, \ldots, & 2p_{n-1,n}, & p_{nn} \end{bmatrix}^T. \tag{5}$$

We denote the dimension of the vector $v(P)$ as $\underline{n} \triangleq \frac{n(n+1)}{2}$. Given vectors $x, y \in \mathbb{R}^n$, define

$$\mathcal{B}(x, y) = \frac{1}{2} \begin{bmatrix} 2x_1 y_1, & x_1 y_2 + x_2 y_1, \ldots, & x_1 y_n + x_n y_1, & 2x_2 y_2, \ldots, & 2x_n y_n \end{bmatrix}^T \in \mathbb{R}^{\underline{n}}. \tag{6}$$

Further properties of these operations can be found in Proposition A.1 of Appendix A.

**RCI Algorithm.** Leveraging Kleinman's structure, RCI uses state-action trajectory data $(x, u)$ to iteratively solve for the optimal policy of the nonlinear system (1).

**Critic Network Structure.** The critic is given by $V(x) = \mathcal{B}^T(x, x)c_i$, where $c_i \in \mathbb{R}^{\underline{n}}$ are the weights yielded from RCI learning (discussed shortly). This neural network approximation structure

for the critic is standard in SOTA ADP methods. Given $c_i \in \mathbb{R}^{\underline{n}}$ and from Proposition A.1, we have that there exists a unique $P_i = P_i^T \in \mathbb{R}^{n \times n}$ such that $c_i = v(P_i)$, and that $V(x) = \mathcal{B}^T(x, x)c_i = \mathcal{B}^T(x, x)v(P_i) = x^T P_i x$, the same quadratic approximation form of Kleinman's algorithm.

**Algorithm Procedure: Overview.** Given a designer-selected state-action sample count $l \in \mathbb{N}$, sequence of sample instants $\{t_k\}_{k=0}^l$, and reference input $r(t)$ (see discussions around Equation (13)), we apply $r(t)$ to the nonlinear system (1) simulating the environment under an initial stabilizing policy $K_0$ and collecting state-action data $\{(x(t_k), u(t_k))\}_{k=0}^l$. With this trajectory data, we use the method of integral reinforcement (Vrabie & Lewis, 2009) to construct a learning weight update:

$$\mathbf{A}_i c_i = \mathbf{b}_i. \tag{7}$$

Here, the learning matrices $\mathbf{A}_i \in \mathbb{R}^{l \times \underline{n}}, \mathbf{b}_i \in \mathbb{R}^l$ contain environment data pertaining to 1) trajectory sample difference data $x(t_k) - x(t_{k-1})$, 2) trajectory integral data $\int_{t_{k-1}}^{t_k} x d\tau$, and 3) the current policy $K_i$. We describe the exact form of these matrices below:

**Step 1.** Given iteration $i \geq 0$, we use the method of integral reinforcement to construct a learning update for the next iteration controller $K_{i+1}$. Let $t_0 < t_1$ be given. The critic network approximates the cost $J$ (2), which implies that along environment trajectories, we have

$$V(x(t_1)) - V(x(t_0)) = \int_{t_0}^{t_1} x^T Q x + u^T R u \, d\tau. \tag{8}$$

The right-hand-side of (8), called the integral reinforcement signal (Vrabie & Lewis, 2009), requires only state-action data $(x, u)$ from the nonlinear system (1). (8) is exact only when the critic $V$ represents the cost $J$ exactly. The learning goal is to minimize residual approximation error in (8).

**Step 2.** In order to recast (8) in a form amenable to learning, we rearrange the terms in (1) as

$$\dot{x} = \tilde{f}(x) + g(x)u + Ax + BK_i x. \tag{9}$$

Here, the drift term $\tilde{f}(x) \triangleq f(x) - Ax \in \mathbb{R}^n$ fully captures 1) the system nonlinearities, and 2) possible model uncertainties, while $A$, $B$, and $A_i \triangleq A - BK_i$ are known nominal linearization terms. We emphasize that the equation (9) is still exact to the original nonlinear dynamics (1). Differentiating the value function $V$ along system trajectories, we have $V(x(t_1)) - V(x(t_0)) = \int_{t_0}^{t_1} \frac{d}{d\tau} \{V(x)\} \, d\tau$, and using the identification $c_i = v(P_i)$ from Proposition A.1, along the solutions of the nonlinear system (1) we have

$$V(x(t_1)) - V(x(t_0)) = x^T(t_1)P_i x(t_1) - x^T(t_0)P_i x(t_0)$$
$$= \int_{t_0}^{t_1} \left[ 2(\tilde{f}(x) + g(x)u + BK_i x)^T P_i x + x^T (A_i^T P_i + P_i A_i) x \right] d\tau. \tag{10}$$

Now, applying the algebraic identity (19) and rearranging terms, (10) becomes

$$\left[ -2 \int_{t_0}^{t_1} \mathcal{B}(\tilde{f}(x) + g(x)u + BK_i x, x) \, d\tau + \mathcal{B}(x(t_1) + x(t_0), x(t_1) - x(t_0)) \right]^T c_i$$
$$= \left[ \int_{t_0}^{t_1} \mathcal{B}(x, x) d\tau \right]^T v(A_i^T P_i + P_i A_i) = - \left[ \int_{t_0}^{t_1} \mathcal{B}(x, x) d\tau \right]^T v(Q + K_i^T R K_i). \tag{11}$$

The integral reinforcement equation (11) is now of the required form for learning: The terms in brackets $\left[ -2 \int_{t_0}^{t_1} \cdots d\tau + \mathcal{B}(x(t_1) \cdots - x(t_0)) \right]^T c_i$ contain the environment trajectory integral and difference data and will form a single row of the learning matrix $\mathbf{A}_i$ (7), multiplied on the right by the network weight vector $c_i$. Meanwhile, the term in $v(Q + K_i^T R K_i)$ requires only integral state data $x$, learning cost $Q$, $R$, and current policy $K_i$. This will form a single element of the learning vector $\mathbf{b}_i$ (7), establishing an integral reinforcement learning update.

**Step 3.** We now use the integral reinforcement (11) (which comprises a single trajectory sample) to construct the learning matrices $\mathbf{A}_i \in \mathbb{R}^{l \times \underline{n}}, \mathbf{b}_i \in \mathbb{R}^l$ (7) from $l$ such samples. Specifically, applying (11) at the sample instants $\{t_k\}_{k=0}^l$ and manipulating further, the learning matrices $\mathbf{A}_i$ and $\mathbf{b}_i$ are

$$\mathbf{A}_i = -2 \left[ I_{\tilde{f}+gu,x} + I_{x,x} W_i^T \right] + \delta_{x,x}, \qquad \mathbf{b}_i = -I_{x,x} v(Q + K_i^T R K_i). \tag{12}$$

Here $W_i \triangleq W \left( I_n \otimes B K_i \right) W_r^{-1}$, where $\otimes$ is the Kronecker product (Brewer, 1978), and $W$, $W_r^{-1}$ (17) are standard Kronecker terms. $I_{\bullet,x}, \delta_{x,x} \in \mathbb{R}^{l \times \underline{n}}$ (18) are simply "storage" matrices containing integral data $I_{\bullet,x} \leftarrow \int_{t_{k-1}}^{t_k} x d\tau$ and difference data $\delta_{x,x} \leftarrow x(t_k) - x(t_{k-1})$ between successive samples as they appear in the integral reinforcement equation (11) and are discussed below (11).

**Step 4.** Having solved for the critic weights $c_i$ (7), we update the policy as (4): $K_{i+1} = R^{-1} B^T v^{-1}(c_i)$ (Proposition A.1), after which we return to (7) to yield $c_{i+1}$, and so on.

**Remark 2.1 (System Dynamics Required by SOTA CT-RL Works)** Comparing the dynamics required by RCI and deep RL FVIs in Table 1, the additional partial derivative knowledge required by FVIs with respect to the state $x$ and model uncertainty parameters $\theta$ is generally susceptible to modeling error (Khalil, 2002). The ADPs generally require little dynamics, but these methods struggle to synthesize designs (Wallace & Si, 2022).

Table 1: System dynamics required by SOTA CT-RL methods

| Algorithm | | | | System dynamics required | | | |
|---|---|---|---|---|---|---|---|
| RCI | | | | $f, g$ | | | |
| FVIs (Lutter et al., 2023a; 2022) | | | | $f, g, \partial f/\partial x, \partial g/\partial x, \partial f/\partial \theta, \partial g/\partial \theta$ (Remark 2.1) | | | |
| IRL | SPI | RADP | CT-VI | $g$ | $f, g$ | None | None (Remark 2.1) |

Table 2: Environments in SOTA CT-RL evaluations (full details in Appendices D-F)

| Algorithm | System | Order | # Inputs | Source of model parameters |
|---|---|---|---|---|
| RCI | Pendulum | $\longrightarrow$ | $\longrightarrow$ | Identical to FVIs below as benchmark |
| | Jet aircraft (new in CT-RL) | 4 | 2 | Full-scale NASA wind tunnel tests (Soderman & Aiken, 1971) |
| | DDMR ground bot (new in CT-RL) | 4 | 2 | System ID on actual hardware (Mondal et al., 2020; 2019) |
| FVIs | Pendulum | 2 | 1 | Quanser STEM curriculum resources (Quanser, 2018) |
| | Cart pendulum | 4 | 1 | |
| | Furatura pendulum | 4 | 1 | |
| ADPs | Simple academic | 2 | 1 | Non-physical, constructed so optimal solutions known *a priori* |

**Reference Command Input (RCI).** Persistence of excitation (PE) requirements are often invoked in proofs of ADP CT-RL algorithm properties (cf. Remark 2.2). To achieve PE, it is standard practice in ADP to apply a probing noise $d$ to the system (1) in a feedback control of the form $u = \mu(x) + d$, where $\mu : \mathbb{R}^n \to \mathbb{R}^m$ is a stabilizing policy. Since good feedback control attenuates plant input disturbances, plant-input probing noise excitation is an inherently problematic practice (Rodriguez, 2004) which was shown a limitation (Wallace & Si, 2022). We propose a reference command input (RCI) solution which instead excites the closed-loop system at the favorable reference command input $r$. Critically, RCI is compatible with current RL formulations. Since full state information is required in the optimal control problem, we may designate a subset of the state $x$ as measurement variables $y \in \mathbb{R}^p$ for reference injection. Writing $x = \begin{bmatrix} y^T & x_r^T \end{bmatrix}^T$, where $x_r \in \mathbb{R}^{n-p}$ denotes the rest of the state, and denoting $e = r - y$ as the tracking error signal, the control

$$u = \mu(e, x_r) = \mu(y, x_r) + \tilde{d}, \qquad \tilde{d} \triangleq \mu(e, x_r) - \mu(y, x_r), \qquad (13)$$

is of the required form $u = \mu(x) + \tilde{d}$. Thus, RCI can improve learning of existing CT-RL algorithms.

We are now ready to present our main theoretical guarantees:

**Theorem 2.1 (Convergence, Optimality, and Closed-Loop Stability of RCI)** Suppose that the initial policy $K_0$ stabilizes the nominal linearization (i.e., that $A - B K_0$ is Hurwitz), and that the sample instants $\{t_k\}_{k=0}^l$ are such that the integral reinforcement matrix $I_{x,x}$ (18) has full rank $\underline{n}$. Identifying $c_i = v(P_i)$ (Proposition A.1), RCI and Kleinman algorithm's ALE (3) produce identical sequences of matrices $\{P_i\}_{i=0}^\infty$ and policies $\{K_i\}_{i=1}^\infty$. As a result, the 1) convergence, 2) solution optimality, and 3) closed-loop stability results of Kleinman's algorithm (Theorem B.1) hold for RCI in the choice of critic bases $\mathcal{B}(x, x) \in \mathbb{R}^{\underline{n}}$ on the nonlinear system (1).
*Proof:* An induction argument presented in Appendix B. ∎

**Remark 2.2 (Theoretical Results and Assumptions of SOTA CT-RL Works)** RCI, like the leading ADP-based works (Vrabie & Lewis, 2009; Vamvoudakis & Lewis, 2010; Jiang & Jiang, 2014; Bian & Jiang, 2022) provides theoretical results of convergence, optimality, and closed-loop stability. The SOTA FVIs (Lutter et al., 2023a; 2022) do not provide these guarantees. Meanwhile, RCI's assumptions are among the least stringent in CT-RL, which we outline in detail in Remark C.3 of Appendix C. RCI requires standard stabilizability, detectability, and full-rank assumptions for well-posedness (see above). Meanwhile, DRL FVIs (Lutter et al., 2023a; 2022) require partial derivatives of $f$ and $g$ (cf. Remark 2.1) and neglect higher-order terms in the Taylor expansion of the optimal value $V^*$. The ADP methods generally have the most stringent assumptions. For instance, CT-VI (Bian & Jiang, 2022) requires PE, existence and uniqueness of solutions to an uncountable family of finite-horizon HJB equations, and an initial globally asymptotically stabilizing policy.

**Remark 2.3 (Decentralizable Environment for Further Data Efficiency)** Consider a decentralized environment $(f, g)$ (1) with $N$ separable control loops. To illustrate, we present $N = 2$ loops:

$$\begin{bmatrix} \dot{x}_1 \\ \dot{x}_2 \end{bmatrix} = \begin{bmatrix} f_1(x) \\ f_2(x) \end{bmatrix} + \begin{bmatrix} g_{11}(x) & g_{12}(x) \\ g_{21}(x) & g_{22}(x) \end{bmatrix} \begin{bmatrix} u_1 \\ u_2 \end{bmatrix}. \tag{14}$$

No assumptions are made on dynamic coupling between the loops; i.e., the loops may be fully coupled. Here, $x_j \in \mathbb{R}^{n_j}$, $u_j \in \mathbb{R}^{m_j}$ $(j = 1, \ldots, N)$ with $\sum_{j=1}^{N} n_j = n$ and $\sum_{j=1}^{N} m_j = m$. Such partitions appear in a variety of real-world applications such as robotic systems (Craig, 2005; Dhaouadi & Abu Hatab, 2013), helicopters (Enns & Si, 2002; 2003b;a), UAVs (Wang et al., 2016), and aircraft (Stengel, 2022; Dickeson et al., 2009a;b) In this case, the RCI learning rule (7) occurs in a decentralized fashion in each of the loops, thereby reducing problem dimensionality. This results in sequences of critic network weights $\{c_{i,j}\}_{i=0}^{\infty}$ and policies $\{K_{i,j}\}_{i=1}^{\infty}$ in each loop yielded by learning analogous to (7), now constructed with $x_j$ instead of $x$, $u_j$ instead of $u$, etc. Crucially, learning convergence, optimality, and closed-loop stability results analogous to Theorem 2.1 hold for the policies $\{K_{i,j}\}_{i=1}^{\infty}$ in each loop.

## 3 EXPERIMENT SETUP FOR EVALUATIONS

We conduct ablation studies with comparisons to baseline LQR, demonstration of robustness and generalization of RCI, and comparisons with the SOTA FVIs (Lutter et al., 2023a; 2022). Performance criteria are defined in Section 3.2.

### 3.1 SELECTION OF SOTA ENVIRONMENTS

Extensive evaluations of RCI are performed on three SOTA environments described in Table 2. The dynamics of the environments are given by

$$\begin{aligned} \dot{\theta} &= \omega \\ \dot{\omega} &= \frac{mgL}{2I}\sin\theta + \frac{\tau}{I} \end{aligned} \quad \left| \quad \begin{aligned} \dot{V} &= \frac{m_c d}{\hat{m}}\omega^2 - \frac{2\overline{\beta}}{\hat{m}r^2}V + \frac{k_t}{\hat{m}k_g r}i_{a_r} + \frac{k_t}{\hat{m}k_g r}i_{a_l} \\ \dot{\omega} &= \frac{-m_c d}{\hat{I}}\omega V - \frac{\overline{\beta}d_w^2}{2\hat{I}r^2}\omega + \frac{d_w k_t}{2\hat{I}k_g r}i_{a_r} - \frac{d_w k_t}{2\hat{I}k_g r}i_{a_l} \\ \dot{i}_{a_r} &= \frac{-k_g k_b}{l_a r}V - \frac{k_g k_b d_w}{2l_a r}\omega - \frac{r_a}{l_a}i_{a_r} + \frac{1}{2l_a}\overline{e}_a + \frac{1}{2l_a}\Delta e_a \\ \dot{i}_{a_l} &= \frac{-k_g k_b}{l_a r}V + \frac{k_g k_b d_w}{2l_a r}\omega - \frac{r_a}{l_a}i_{a_l} + \frac{1}{2l_a}\overline{e}_a - \frac{1}{2l_a}\Delta e_a \end{aligned} \right. \tag{15}$$

$$\begin{bmatrix} \dot{V} \\ \dot{\gamma} \\ \dot{q} \\ \dot{\alpha} \end{bmatrix} = \begin{bmatrix} -D_V & -g\cos\alpha_e & 0 & 0 \\ \frac{L_V}{V_e} & 0 & 0 & \frac{L_\alpha}{V_e} \\ 0 & 0 & M_q & M_\alpha \\ \frac{-L_V}{V_e} & 0 & 1 & \frac{-L_\alpha}{V_e} \end{bmatrix} \begin{bmatrix} V \\ \gamma \\ q \\ \alpha \end{bmatrix} + \begin{bmatrix} T_{\delta_T} & 0 \\ 0 & 0 \\ 0 & M_{\delta_E} \\ 0 & 0 \end{bmatrix} \begin{bmatrix} \delta_T \\ \delta_E \end{bmatrix}. \tag{16}$$

**The pendulum** (15, left) has states $x = (\theta, \omega)$, where $\theta$ is the pendulum angle (measured zero pointing upward, positive counterclockwise), $\omega$ is the pendulum angular velocity, and the single-input control $u = \tau$ is the torque applied to the pendulum base. **The jet** (16) has states $x = (V, \gamma, q, \alpha)$, where $V$ is the airspeed, $\gamma$ is the flightpath angle (FPA), $q$ is the pitch rate, and $\alpha$ is the angle of attack (AOA). It has controls $u = (\delta_T, \delta_E)$, where $\delta_T$ is the throttle setting (associated with the airspeed $V$ in the translational loop $j = 1$), and $\delta_E$ is the elevator deflection (associated with the FPA $\gamma$ and attitude $q, \alpha$ in the rotational loop $j = 2$). The differential drive mobile robot (**DDMR**) (15, right) also lends itself to decentralization, with states $x = (V, \omega, i_{a_r}, i_{a_l})$, where $V$ is

the velocity, $\omega$ is the angular velocity, and $i_{a_r}, i_{a_l}$ are the right and left DC motor armature currents, respectively. The controls are $u = (\overline{e}_a, \Delta e_a)$, where $\overline{e}_a$ is the average of the armature voltages applied to the right and left DC motors (associated with the speed $V$ in the translational loop $j = 1$), and $\Delta e_a$ is the difference of the right/left voltages (associated with the rotational velocity $\omega$ in the rotational loop $j = 2$).

## 3.2 IMPLEMENTATION AND TRAINING PROCEDURES

**All Code/Data Available.** All RCI code and datasets for this study are available in supplemental and at Anonymized (2024). All FVI results (Lutter et al., 2023a; 2022) are generated by the open-source code developed by the authors available at Lutter et al. (2023b). **Hyperparameters: RCI.** Can be found in Table 16 of Appendix G. 1) Reference command inputs $r$: After in-depth analysis of the system physics in Appendices D-F, we choose the references $r$ based on the natural input/output resonances of each environment to maximize excitation efficiency. 2) Sample period $T_s = t_k - t_{k-1}$: chosen based on the natural bandwidth of each environment. 3) Number of samples $l$: chosen based on the system dimensionality $n$. 4) Number of iterations $i^*$: 5 iterations was observed to be sufficient for convergence of all environments, a low number due to RCI data efficiency. **Hyperparameters: FVIs.** Can be found in Table 17 of Appendix G. For the pendulum system, we use the FVI setup/hyperparameters of the original studies (Lutter et al., 2023a; 2022) with a few minor implementation changes to make results comparable among the methods (cf. Appendix G for discussion). For the jet and DDMR examples that are new to FVIs, we have chosen hyperparameters in light of the successes achieved by the selections in Lutter et al. (2023a; 2022), tailored to maximize learning performance for these specific systems. **Generalizations.** To evaluate generalization to model uncertainty, we account for different levels of modeling error $\nu \in \mathbb{R}$ for each system over a grid of values $\nu \in G_\nu$ (cf. Appendix G for selections), ranging from zero modeling error ($\nu = 1$) to a 25% modeling error ($\nu = 1.25$ for the pendulum and DDMR, $\nu = 0.75$ for the jet). The direction of the perturbation (i.e., $\nu > 1$ or $\nu < 1$) is chosen to maximize the difficulty of the learning problem (cf. Appendices D-F for in-depth discussion). **Performance Measures.** To quantify learning performance, we use the following measures: 1) policy cost performance $J$ (2), 2) critic network approximation error $J - V$, 3) closed-loop time-domain responses, 4) algorithm data/time efficiency and number of free parameters, and 5) generalization to model uncertainty $\nu$. **Hardware, Software.** These studies are performed in MATLAB R2022b, on an NVIDIA RTX 2060, Intel i7 (9th Gen) processor. Integrations are performed in MATLAB's adaptive ode45 solver. For an in-depth discussion of our setup, and for a complete list of numerical hyperparameter selections, see Appendix G.

**Training: RCI.** Online data is collected from the actual system (i.e., with modeling error $\nu \neq 1$) over one simulation beginning at an initial condition $x_0$. We thus define learning over a pair $(x_0, \nu)$ as constituting one *trial* for RCI. We present data for RCI trained over the trial space $(x_0, \nu) \in G_{x_0} \times G_\nu$ (cf. Appendix G for numerical selections). This corresponds to 1,620 trials for the pendulum (81 ICs $x_0 \times$ 20 modeling errors $\nu$), 1,089 trials for the jet (99 $x_0 \times$ 11 $\nu$), and 1,287 trials for the DDMR (117 $x_0 \times$ 11 $\nu$). **Training: FVIs.** The notion of a trial in FVI training differs from that of RCI. To execute to completion, FVI requires training data from over 5 million simulations (cf. Table 9) initialized in a uniform distribution $\mathcal{U}$ over the state domain, for details see Lutter et al. (2023a; 2022). Thus, for FVI a *trial* is associated with the specific random number generation seed. In this work, we evaluated the FVI algorithms over 20 seeds for each environment. All surface plot results (e.g., Figure 2) correspond to seed 42, the same as the original works (Lutter et al., 2023a; 2022). For further discussion of FVIs training, see Appendix C.

## 4 ABLATION STUDY

In this evaluation, we provide quantitative assessment of the nonlinear RCI algorithm over the baseline classical LQR design performed on the nominal linearization $(A, B)$ (Section 2).

**RCI on Nominal/Perfect Model $\nu = 1$ and Ablation Sweeps of Initial Conditions (ICs).** A complete evaluation of RCI over a systematic sweep of ICs and modeling errors (different $\nu$) for all three environments can be found in Tables 18-20 of Appendix H. For illustration, some of this data is in Table 3 for the pendulum. At *worst-case* over the IC grid $x_0 \in G_{x_0}$, on the nominal model $\nu = 1$ RCI converges to within $3.71 \times 10^{-5}$ of the optimal policy $K^*$ for the pendulum (Table 3), $2.42 \times 10^{-8}$ for the jet, and $5.17 \times 10^{-5}$ for the DDMR. Thus, RCI achieves real-world convergence

performance in accordance with its theoretical guarantees. As a comparison, the FVIs also converge nicely, but they still exhibit appreciable variance between seeds (cf. Figure 1).

Table 3: RCI initial condition $x_0$ and modeling error $\nu$ ablation on pendulum

| $\nu$ (24) | Nom LQ policy optimality error $\|K_0 - K^*\|$ | Data over $x_0 \in G_{x_0}$ | RCI policy optimality error $\|K_{i^*} - K^*\|$ | Percent reduction Nom LQ $\rightarrow$ RCI |
|---|---|---|---|---|
| 1 | 0 (Nom LQ optimal) | worst | 3.71e-05 | N/A |
| | | avg±std | 1.75e-05±1.19e-05 | N/A |
| 1.1 | 1.04 | worst | 0.06 | 94.12 |
| | | avg±std | 0.03±5.47e-03 | 96.75±0.53 |
| 1.25 | 2.61 | worst | 0.20 | 92.18 |
| | | avg±std | 0.13±0.01 | 95.21±0.54 |

**RCI Learning Generalization in the Presence of Modeling Error $\nu \neq 1$.** Tables 18-20 of Appendix H summarize the optimality errors $\|K_{i^*} - K^*\|$ between the RCI final policies $K_{i^*}$ and the optimal policy $K^*$, reproduced here for the pendulum in Table 3. As a worst case observed over the IC grid $x_0 \in G_{x_0}$ and in the presence of the most severe 25% modeling error tested, RCI converges to within 0.20, $1.78 \times 10^{-8}$, and 0.32 of the optimal policy $K^*$ for the pendulum, jet, and DDMR, respectively. By comparison, the optimality errors of the nominal LQ controllers $K_0$ are 2.61, 0.13, and 1.74, respectively. Thus, for the pendulum we see that RCI's policy error $\|K_{i^*} - K^*\|$ is *at most* 0.20/2.61 (8%) the policy error of the nominal LQ design $\|K_0 - K^*\|$; i.e., a reduction of *at least* 92% for a given initial condition $x_0 \in G_{x_0}$. Similarly, RCI offers a reduction of at least 99.99% and 81% from the nominal LQ design for the jet and DDMR, respectively, thus demonstrating generalizability of learning with respect to environment errors.

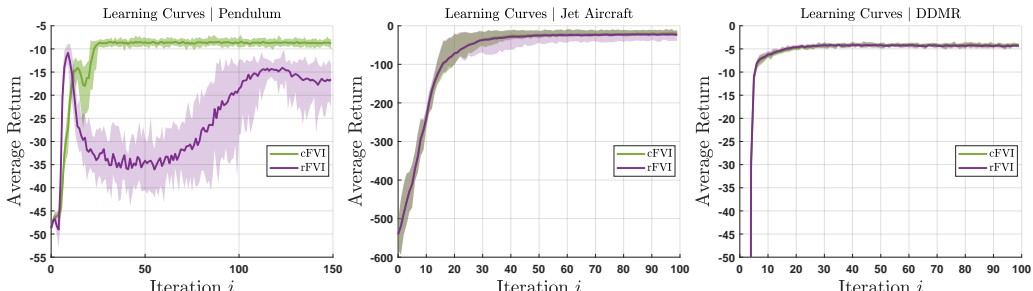

Figure 1: Learning curves of cFVI and rFVI obtained over 20 seeds for the pendulum (left), jet aircraft (middle), and DDMR (right). The shaded area displays the min/max range between seeds, as is presented in the original works (Lutter et al., 2023a; 2022).

## 5 QUANTITATIVE COMPARISONS BETWEEN RCI AND DEEP RL FVIs

**FVIs as Benchmark.** We plot the cFVI and rFVI learning curves for all three systems in Figure 1. As can be seen, these algorithms exhibit overall consistent learning behavior as shown in the original works (Lutter et al., 2023a; 2022), a result confirmed independently here on SOTA environments.

**Cost Performance.** Figure 2 illustrates the cost difference data of cFVI and rFVI with respect to RCI, summarized in Table 4. Note that wherever this difference is positive, RCI delivers *better* performance than the respective FVI algorithm. Several key trends emerge from Table 4: 1) RCI achieves the lowest cost for all three systems as modeling error $\nu$ is increased, and the best modeling error generalization overall. 2) For both multi-loop systems (i.e., the jet and DDMR), RCI achieves lowest cost pointwise, regardless of modeling error. 3) The FVIs perform quite well. Indeed, the leftmost plot in Figure 2 shows that cFVI performance edges out that of RCI for the nominal pendulum far from the origin $x = 0$. However, when modeling error is introduced (second from the left), cFVI performance degrades significantly, a trend we observe for all three systems (cf. Appendix I). By contrast, rFVI degradation is less pronounced, but at the cost of inferior overall performance.

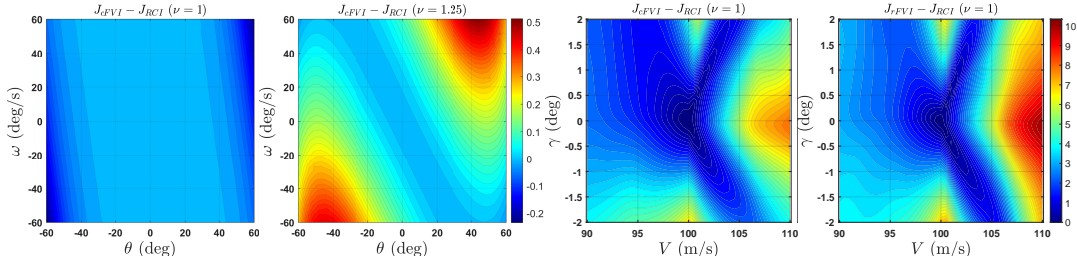

Figure 2: Left two figures: cFVI cost difference $J_{cFVI} - J_{RCI}$ for the nominal pendulum model $\nu = 1$ (first) and at 25% modeling error $\nu = 1.25$ (second). Right two figures: cFVI (third) and rFVI (fourth) cost difference for the nominal jet aircraft model $\nu = 1$. Full plots in Appendix I.

Table 4: Training cost difference data $J_{xFVI} - J_{RCI}$ ($> 0$: RCI better)

| Alg | Data | Pendulum | | Jet aircraft | | DDMR | |
|-----|------|----------|-----------|--------------|--------------|----------|----------|
| | | $\nu = 1$ | $\nu = 1.25$ | $\nu = 1$ | $\nu = 0.75$ | $\nu = 1$ | $\nu = 1.25$ |
| cFVI | min | -0.23 | -0.02 | 0.00 | 0.00 | 1.11e-05 | 1.08e-05 |
| | max | 4.62e-04 | 0.51 | 8.04 | 8.57 | 0.27 | 0.41 |
| | avg | -0.02±0.04 | 0.12±0.12 | 3.08±1.80 | 3.74±1.98 | 0.09±0.08 | 0.13±0.11 |
| rFVI | min | -1.33e-06 | -3.81e-04 | 0.00 | 0.00 | 1.77e-03 | 1.77e-03 |
| | max | 7.72 | 10.27 | 10.15 | 10.34 | 2.27 | 1.60 |
| | avg | 2.16±1.92 | 2.99±2.61 | 3.72±2.23 | 4.22±2.29 | 0.63±0.54 | 0.46±0.38 |

**Estimation Error.** Figure 3 shows the critic network error $J - V$ for the three methods on the DDMR at 25% modeling error. As is the case with cost performance, RCI exhbits the smallest critic network error when modeling error is introduced and the best generalization overall. cFVI does an excellent job of approximating its policy cost for the nominal model but experiences significant degradation. Indeed, Table 23 in Appendix I shows that cFVI's worst-case critic error increases by 454% from $\nu = 1$ to $\nu = 1.25$, as compared to RCI's 39%. Meanwhile, rFVI struggles with cost approximation to a larger degree than RCI or cFVI; however, rFVI's worst-case cost approximation improves from 6.08 at nominal to 5.36 at 25% modeling error (cf. Table 23), demonstrating favorable generalization.

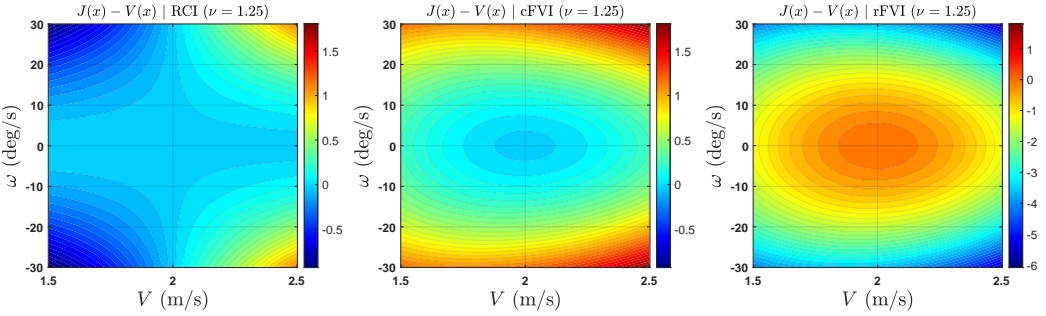

Figure 3: Critic NN approximation error $J(x) - V(x)$ for DDMR environment at 25% modeling error $\nu = 1.25$ (43). Left: RCI, middle: cFVI, right: rFVI. Note: rFVI color normalized independently for legibility purposes. Full plots can be found in Appendix I.

**Closed-Loop Performance.** Figure 4 displays closed-loop responses for all three systems at 25% modeling error. Overall, the FVI responses are either sluggish and/or exhibit large overshoot when compared to RCI. As corroborated by Section 4, RCI recovers the closed-loop performance of the optimal policy for all systems and outperforms the nominal classical LQ design.

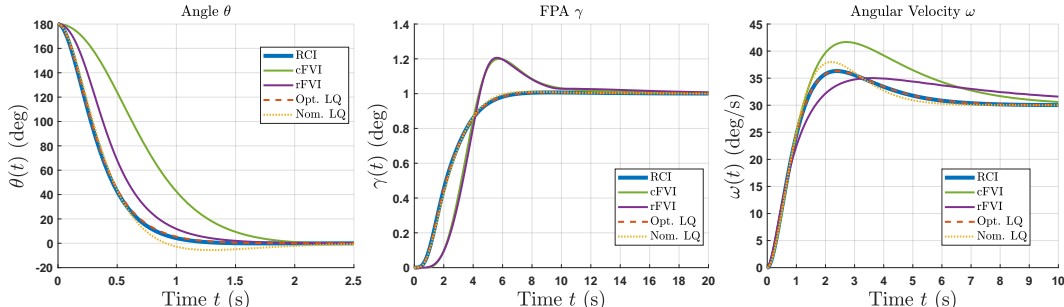

Figure 4: Closed-loop responses at 25% modeling error. Left: pendulum swing-up from natural hanging position (i.e., $\theta = 180°$). Middle: jet aircraft 1 deg step FPA command. Right: DDMR 30 deg/s step angular velocity command. Full plots can be found in Appendix I.

**Algorithm Time/Data/Parameter Complexity.** Refer to Table 9 of Appendix C, which lists key algorithm complexity parameters for RCI and FVI. On the DDMR, for example, the ratio of RCI/FVI for simulations required: 1/5,000,000, data samples: 1/6,000,000, network weights: 1/12,000, training epochs: 1/400, number of hyperparameters: 1/4, and training time: 1/3,000. As a result, we are able to conduct 160 times the number of learning trials for RCI in these studies than for FVI.

## 6  CONCLUSION AND DISCUSSION

In the context of current ADP and deep RL CT-RL methods, we formulate a model-based RCI algorithm which leverages nonlinear learning alongside input/output insights of the environment and Kleinman control structures for data efficiency. RCI leads to new CT-RL results on SOTA environments (jet aircraft and DDMR ground robot new to CT-RL). All three CT-RL classes represent different approaches to the learning control problem. **RCI** presents theoretical guarantees, and its learning performance at least matches, and often outperforms, the SOTA deep RL FVIs in terms of 1) policy cost performance, 2) critic network approximation performance, 3) closed-loop time-domain performance, 4) algorithm data/time efficiency, and 5) generalization to modeling error. Yet, RCI's efficiency requires knowledge of the environment, and RCI only considers Q-R cost structures (2). Meanwhile, **ADP** presents strong analytical results and may not require knowledge of the environment $(f, g)$, but these methods have not been proven for meaningful applications, as only evaluations of simple systems with known optimal solutions are available. Furthermore, ADPs generally restrict to Q-R cost as well. Finally, deep RL **FVIs** are learning-driven methods with significant empirical promise and generalizability, as independently verified by the new SOTA evaluations we conduct on these algorithms in Section 5. These methods also consider flexible cost structures including dense/sparse costs. However, FVIs require the most dynamic knowledge of the three classes, and theoretical results are yet to be developed.

**Limitations of this Study.** Our reference command as probing signal requires understanding of the environment physics, which may be restrictive when the model is unknown. Furthermore, for data efficiency we have restricted RCI to Q-R cost (2). While Q-R cost addresses a variety of control problems in CT-RL, RCI does not have the flexibility to consider other forms of the cost structure.

**Reproducibility Statement.** All RCI code and all datasets for this study are available in supplemental and at Anonymized (2024). All FVI results (Lutter et al., 2023a; 2022) are generated by the open-source code developed by the authors available at Lutter et al. (2023b). For an in-depth discussion of our setup and a complete list of numerical hyperparameter selections, see Section 3.2 and Appendix G. All theoretical assumptions can be found in Remark 2.2, and all proofs of theoretical results can be found in Appendix B.

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

## APPENDIX A   RELEVANT OPERATORS FOR RCI ALGORITHM

**Definition A.1 (Relevant Operators for Learning)** Let $\underline{n} \triangleq \frac{n(n+1)}{2}$, and define the maps $v :$ $\mathbb{R}^{n \times n} \to \mathbb{R}^{\underline{n}}$, and $\mathcal{B} : \mathbb{R}^n \times \mathbb{R}^n \to \mathbb{R}^{\underline{n}}$ as in (5) and (6), respectively. Define $W \in \mathbb{R}^{\underline{n} \times n^2}$ (and its right inverse $W_r^{-1} \in \mathbb{R}^{n^2 \times \underline{n}}$) as the matrices satisfying the identities

$$\mathcal{B}(x,y) = W(x \otimes y), \qquad x \otimes x = W_r^{-1} \mathcal{B}(x,x), \qquad \forall\, x, y \in \mathbb{R}^n, \tag{17}$$

where $\otimes$ is the Kronecker product (Brewer, 1978). For $l \in \mathbb{N}$ and a strictly increasing sequence $\{t_k\}_{k=0}^l$, whenever $x, y : [t_0, t_l] \to \mathbb{R}^n$, define the matrices $\delta_{x,y} \in \mathbb{R}^{l \times \underline{n}}$ and $I_{x,y} \in \mathbb{R}^{l \times \underline{n}}$ as

$$\delta_{x,y} = \begin{bmatrix} \mathcal{B}^T\big(x(t_1) + y(t_0), x(t_1) - y(t_0)\big) \\ \vdots \\ \mathcal{B}^T\big(x(t_l) + y(t_{l-1}), x(t_l) - y(t_{l-1})\big) \end{bmatrix}, \quad I_{x,y} = \begin{bmatrix} \int_{t_0}^{t_1} \mathcal{B}^T(x,y)\, d\tau \\ \vdots \\ \int_{t_{l-1}}^{t_l} \mathcal{B}^T(x,y)\, d\tau \end{bmatrix}. \tag{18}$$

**Proposition A.1** The operators $v$ (5), $\mathcal{B}$ (6), and matrices $W, W_r^{-1}$ (17) have the following properties:

(i) The restriction of $v$ to the symmetric matrices is a linear isomorphism, and $\mathcal{B}$ is a symmetric bilinear form.

(ii) Whenever $P \in \mathbb{R}^{n \times n}$, $P = P^T$, the following identity holds

$$\mathcal{B}^T(x,y)v(P) = x^T P y, \quad \forall\, x, y \in \mathbb{R}^n. \tag{19}$$

(iii) The matrices $W, W_r^{-1}$ are uniquely determined the identity (17).

## APPENDIX B   PROOFS OF THEORETICAL RESULTS

The following theorem will be needed to prove the main result. Here, let $K^* = R^{-1}B^T P^*$ denote the optimal LQR controller, where $P^* \in \mathbb{R}^{n \times n}$, $P^* = P^{*T} > 0$ is the solution of the Riccati equation (Rodriguez, 2004).

**Theorem B.1 (Convergence, Optimality, and Closed-Loop Stability of Kleinman's Algorithm (Kleinman, 1968))** Suppose the initial policy $K_0$ is such that $A - BK_0$ is Hurwitz. Then we have the following:

(i) $A - BK_i$ is Hurwitz for all $i \geq 0$.

(ii) $P^* \leq P_{i+1} \leq P_i$ for all $i \geq 0$, and $\lim_{i \to \infty} P_i = P^*$, $\lim_{i \to \infty} K_i = K^*$.

We are now ready to proceed with the proof of Theorem 2.1:

### B.1   PROOF OF THEOREM 2.1

Suppose that $A_0 = A - BK_0$ (9) is Hurwitz, and the sample count $l \in \mathbb{N}$ and sample times $\{t_k\}_{k=0}^l$ are such that $I_{x,x}$ (18) has full rank $\underline{n}$.

Suppose it has been proved for iteration $i \geq 0$ that $A_i = A - BK_i$ is Hurwitz. We first claim the hypotheses imply that the least-squares matrix $\mathbf{A}_i \in \mathbb{R}^{l \times \underline{n}}$ (7) also has full column rank $\underline{n}$. For suppose $v(P) \in \mathbb{R}^{\underline{n}}$ is such that $\mathbf{A}_i v(P) = 0$. Examining (11), we note for *any* symmetric matrix that $\mathbf{A}_i v(P) = I_{x,x} v(N)$, where $N \in \mathbb{R}^{n \times n}$, $N = N^T$ is

$$N = A_i^T P + P A_i. \tag{20}$$

However, (20) is itself an ALE. Furthermore, since $N = N^T$ and since $A_i = A - BK_i$ is Hurwitz by hypothesis, (20) has the unique solution $P = \int_0^\infty e^{A_i^T t}(-N)e^{A_i t}\, dt$ (Rodriguez, 2004). Meanwhile, the full rank of $I_{x,x}$ and that $I_{x,x} v(N) = 0$ imply $v(N) = 0$, or $N = 0$. Since $N = 0$, we have by

the above that $v(P) = 0$. We have shown that $\mathbf{A}_i$ has trivial right null space, hence full column rank $\underline{n}$.

Having established that $\mathbf{A}_i$ has full rank, we now claim that $P_i \in \mathbb{R}^{n \times n}$, $P_i = P_i^T > 0$ (uniquely) solves the ALE (3) if and only if $c_i = v(P_i)$ satisfies the least-squares regression (7) at equality. The forward direction was already proved in the derivation (10), (11). Conversely, suppose $v(P) \in \mathbb{R}^n$ is such that the least-squares regression (7) is minimized. Since $\mathbf{A}_i$ has full column rank, $v(P) \in \mathbb{R}^{\underline{n}}$ is unique. Now, letting $P_i = P_i^T > 0$ be the (unique) solution of the ALE (3), (10),(11) establish that $v(P_i) \in \mathbb{R}^{\underline{n}}$ satisfies (7) at equality. Thus, $v(P) = v(P_i)$, whence $P = P_i$ (Proposition A.1) and the result is proved.

Having established the preceding, the proof now follows by induction on the algorithm iteration $i$. ∎

## APPENDIX C    RCI COMPARED TO SOTA CT-RL RESULTS IN DEEP RL AND ADP

In this section, we illustrate a holistic overview of the qualitative and quantitative differences between RCI and the leading CT-RL methods in ADP (Vrabie & Lewis, 2009; Vamvoudakis & Lewis, 2010; Jiang & Jiang, 2014; Bian & Jiang, 2022), and in deep RL FVI (Lutter et al., 2023a; 2022). Comprehensive comparisons show that the proposed RCI method is SOTA.

**Remark C.1 (Environments Studied by SOTA CT-RL Works)** We provide an overview of the environments studied in the evaluations of the leading CT-RL works in Table 5 below. As can be seen the proposed environments are SOTA.

Table 5: Environments in SOTA CT-RL evaluations (full details in Appendices D-F)

| Algorithm | System | Order | # inputs | Source of model parameters |
|---|---|---|---|---|
| RCI | Pendulum | $\longrightarrow$ | $\longrightarrow$ | Identical to FVIs below as benchmark |
| | Jet Aircraft (new in CT-RL) | 4 | 2 | Full-scale NASA wind tunnel tests (Soderman & Aiken, 1971) |
| | DDMR (new in CT-RL) | 4 | 2 | System ID on actual hardware (Mondal et al., 2020; 2019) |
| FVIs | Pendulum | 2 | 1 | Quanser STEM curriculum resources |
| | Cart Pendulum | 4 | 1 | (Quanser, 2018) |
| | Furatura Pendulum | 4 | 1 | |
| IRL | Simple Academic | 2 | 1 | Non-physical, constructed so optimal |
| | Simple Academic | 2 | 1 | solutions known *a priori* (Remark 1) |
| SPI | Simple Linear | 3 | 1 | Non-physical LQR example |
| | Simple Academic | 2 | 1 | See IRL above |
| RADP | Simplified Engine | 2 | 1 | Non-physical, |
| | Simplified Power Bus | 2 | 1 | chosen for illustration |
| CT-VI | Simple Academic | 2 | 1 | See IRL above |
| | Simplified Robot Arm | 4 | 2 | Non-physical, chosen for illustration |

Remark 1: The leading ADP works (Vrabie & Lewis, 2009; Vamvoudakis & Lewis, 2010; Jiang & Jiang, 2014; Bian & Jiang, 2022) almost universally study simple academic second-order examples which are constructed such that the optimal value and policy are polynomial functions known *a priori* in closed form, and for which the bases chosen can achieve exact approximation. Bian & Jiang (2022) do study a robotic arm example, but the model is highly simplified and the parameter values are chosen academic for illustration.

**Remark C.2 (Data and Dynamical Information Required by SOTA CT-RL Works)** We provide an overview of the system data and environment dynamical information required of the leading CT-RL works in Table 6 below.

Table 6: Data and system dynamics required by SOTA CT-RL methods

| Algorithm | Data | System dynamics required |
|---|---|---|
| RCI | $(x, u)$ | $f, g$ |
| cFVI (Lutter et al., 2023a) | $(x, u)$ | $f, g, \partial f/\partial x, \partial g/\partial x$ (Remark 1) |
| rFVI (Lutter et al., 2022) | $(x, u)$ | $f, g, \partial f/\partial x, \partial g/\partial x, \partial f/\partial \theta, \partial g/\partial \theta$ (Remark 1) |
| IRL (Vrabie & Lewis, 2009) | $(x, u)$ | $g$ |
| SPI (Vamvoudakis & Lewis, 2010) | $(x, u)$ | $f, g$ |
| RADP (Jiang & Jiang, 2014) | $(x, u)$ | None (Remark 2) |
| CT-VI (Bian & Jiang, 2022) | $(x, u)$ | None (Remark 2) |

Remark 1: Note that the deep RL FVIs (Lutter et al., 2023a; 2022) require more dynamics knowledge than RCI, in particular partial derivative knowledge with respect to the state $x$ and model uncertainty parameters $\theta$, which is generally highly susceptible to modeling error (Khalil, 2002).

Remark 2: RCI and FVIs are edged out by the RADP (Jiang & Jiang, 2014) and CT-VI (Bian & Jiang, 2022) in dynamical information required, but at the cost of highly restrictive theoretical assumptions (cf. Remark C.3) and significant empirical issues (Wallace & Si, 2022).

**Remark C.3 (Theoretical Assumptions Required by SOTA CT-RL Works)** As shown below, RCI is among the least restrictive in CT-RL in its theoretical assumptions. As a note, all methods require that be Lipschitz near origin to assure well-posedness of solutions to the system differential equations.

- RCI (present work):

    - Stabilizability of the linearization $(A, B)$ of the nonlinear system $(f, g)$ (1) (for well-posedness of regulation problem)
    - $(Q^{1/2}, A)$ detectable (for definiteness of underlying ALE solution)
    - Full column rank of integral reinforcement matrix $I_{x,x}$ (18)
    - Initial stabilizing linear controller $K_0$

- FVIs (Lutter et al., 2023a; 2022):

    - $f$ and $g$ are smooth in their partial derivatives in the state $x$ and model uncertainty parameters $\theta$, and these partials are all known *a priori*
    - Undiscounted problem $\gamma = 1$ can be approximated by discounted problem $0 < \gamma < 1$
    - Discrete-time running cost $r(x, u)$ can be approximated by continuous-time counterpart: $r(x, u) = \Delta t\, r_c(x, u)$ with sample time $\Delta t$
    - Strict convexity of action penalty $g_c$
    - Availability of convex conjugate function to action penalty $g_c$
    - Higher-order terms in Taylor series expansion of optimal value $V^*$ are negligible
    - Existence of an *a priori* state grid $x \in \mathcal{D}$ to contain trajectories to for fitting procedure
    - Trajectories leaving the grid $x \in \mathcal{D}$ can be instantaneously re-initialized to the previous position inside the grid

- IRL (Vrabie & Lewis, 2009):

    - There exists a sequence of sampling instants $t_0 < t_1 < \cdots < t_l$ such that the IRL regression matrix has full rank
    - Chosen basis functions approximate optimal value and its gradient uniformly on compact sets
    - Basis functions for critic network are linearly-independent
    - Initial stabilizing policy

- SPI (Vamvoudakis & Lewis, 2010):

- Existence and uniqueness of least-squares solution to approximate HJB equation
- PE assumption on various learning signals
- Chosen basis functions approximate optimal value and its gradient uniformly on compact sets
- Chosen basis functions approximate optimal policy uniformly on compact sets
- Basis functions for critic network are linearly-independent
- Basis functions for actor network are linearly-independent
- Initial stabilizing policy

- RADP (Jiang & Jiang, 2014):

  - Optimal value can be bounded from above and below by *a priori* known class $\mathcal{K}_\infty$ functions
  - Existence of *a priori* known compact set $\Omega_0$ for which the closed-loop system under the initial policy is invariant with respect to the probing noise $d$
  - PE assumption on various learning signals
  - Chosen basis functions approximate optimal value and its gradient uniformly on compact sets
  - Chosen basis functions approximate optimal policy uniformly on compact sets
  - Basis functions for critic network are linearly-independent
  - Basis functions for actor network are linearly-independent
  - Initial stabilizing policy

- CT-VI (Bian & Jiang, 2022):

  - Existence and uniqueness of solutions to an uncountable family of finite-horizon HJB equations
  - Properness of each solution to the finite-horizon HJB equation
  - Convergence of family of solutions of finite-horizon HJB equation to the infinite-horizon HJB solution
  - Invariance of closed-loop state/action trajectory to compact set with respect to the probing noise $d$
  - Initial *globally asymptotically stabilizing* policy
  - PE assumption on various learning signals
  - Chosen basis functions approximate optimal value and its gradient uniformly on compact sets
  - Chosen basis functions approximate optimal policy uniformly on compact sets
  - Chosen basis functions approximate optimal Hamiltonian uniformly on compact sets
  - Basis functions for critic network are linearly-independent
  - Basis functions for actor network are linearly-independent
  - Basis functions for Hamiltonian network are linearly-independent

**Remark C.4 (RCI vs. FVI: Qualitative Overview)** We present a general overview of these two algorithm types in Table 7. For RCI, the structural simplicity afforded by classical LQ frameworks places little requirements on data. In fact, all state trajectory data required for execution is collected over a single trial under a single stabilizing controller $K_0$. Controller updates are performed after data collection, and equivalence to the classical Kleinman algorithm ensures that each successive controller $K_i$ $(i = 1, 2, \dots)$ is stabilizing (cf. Section 2). By contrast, the deep learning data needs of FVI require too many trials to execute practically in hardware, so data must be collected offline in simulation. FVI's target function fitting requires collection of data under the current-iteration policy. Furthermore, since FVI lacks theoretical stability guarantees, each of these policies need not be stabilizing. Rather, simulations are terminated and re-initialized after a fixed time horizon regardless of the stability of the controller, posing further limitations on adapting this algorithm for training in hardware.

In the case of RCI, a nominal model is required for the regression (7), but its ability to collect data from the actual system allows optimality error reductions (cf. Sections 4, 5). Meanwhile, FVI

Table 7: RCI vs. FVI: Qualitative overview

| Parameter | RCI | cFVI/rFVI |
|---|---|---|
| Underlying Approach | Classical control | Deep RL |
| Offline/Online | Online | Offline |
| Model Required | Yes | Yes |
| Initial Stabilizing Policy Req'd | Yes | No |
| Activation | Quadratic (fixed) | Flexible |
| # NN Layers | 1 (fixed) | Large |
| # Weights/Layer | $\underline{n}$ (fixed) | Large |

requires a nominal model to conduct its simulations, and the lack of practical ability to train its networks from actual system data leaves this algorithm agnostic to any modeling error present in the physical process. In particular, cFVI policy degradation with modeling error is numerically apparent in our evaluations of Section 5. rFVI attempts to learn a more robust policy by its adversarial training structure (Lutter et al., 2022), but as is shown in Section 5 this has consequences on the overall performance of rFVI in relation to cFVI and RCI.

In terms of network design, FVI's use of deep networks offers significant advantages over RCI, which fixes the basis functions as the monomials of degree two $\mathcal{B}(x, x) \in \mathbb{R}^{\underline{n}}$ – the basis associated with the LQR problem. On the other hand, FVI allows the designer to select the network activation (tanh, sigmoid, etc.), enabling potentially greater learning performance based on application-specific needs. Furthermore, RCI uses a single-layer architecture, while FVI network dimensions may also be chosen to fit the needs of the problem. This represents a significant juncture between the two approaches: On one hand, RCI's LQR-based network enables its convergence and closed-loop stability guarantees (cf. Section 2), yet this choice of basis restricts designer flexibility and places approximation capabilities at a disadvantage to FVI's deep networks. Indeed, in evaluation FVI exhibits superior value function approximation on the nominal model, before modeling error gives the approximation advantage to RCI. This network choice has ramifications on the final policies, as well. RCI outputs an LQ controller, which for systems with strong nonlinearities may not achieve the performance of FVI's nonlinear policy network. On the nominal pendulum model in Section 5, for example, FVI's policy offers better performance in regions far from the origin, before modeling error gives RCI the advantage.

**Remark C.5 (Hyperparameter Suites: Greater Design Intuitiveness with RCI)** We now examine Table 8, which lists the hyperparameter suites of RCI and FVI. Per loop $j$, RCI requires less than a fourth the selections of FVI. Even though the number of RCI hyperparameter selections increases linearly with the number of loops, each of these parameters is explicitly linked to the loop dynamics (e.g., sample period $T_{s,j}$ to loop bandwidth, number of samples $l_j \geq \underline{n}_j$ to loop dimension, reference excitation $r_j$ to the closed-loop complementary sensitivity map shapeable via classical control techniques by the initial controller $K_{0,j}$, etc.). Thus, the designer may select learning parameters optimized to the inherent physics of each loop, rather than being forced to select a single set of "middle-ground" parameters for the aggregate system which fails to adequately address individual-loop learning needs. Furthermore, the designer is afforded the luxury of performing troubleshooting at the individual loop level, greatly increasing transparency. By contrast, little intuition is available to select the network hyperparameters of FVI, besides the general rule of thumb that more weights and deeper networks offer better approximation performance. Here, the physics-based dynamical insights offered by RCI's classical underpinning offers clear advantages to designers. In particular, choosing adversary bounds for the rFVI learning which yield improved robustness without disrupting its value function fitting is particularly difficult and offers little systematic insights.

**Remark C.6 (Comparing RCI to FVI (Lutter et al., 2023a; 2022): Significant Reductions in Time/Data Complexity)** We have assembled some key algorithm parameters used in our training studies to gauge the general numerical complexity of RCI in relation to FVI. When compared to FVI, the reductions in algorithm complexity offered by RCI are substantial: On the DDMR, for example, RCI requires 1/5,000,000 the simulations, 1/6,000,000 the data samples, 1/12,000 the weights, 1/400 the epochs, and 1/3,000 the time to train the same model as FVI. These reductions in computational and numerical requirements are a deliberate aim of the RCI algorithm design philosophy, illustrating that dynamical insights and classical control principles may be leveraged to greatly enhance RL learning efficiency and performance. This simplicity comes with further advantages for real-world

Table 8: RCI vs. FVI: learning hyperparameter suites

| RCI loop $j$ | cFVI/rFVI |
|---|---|
| Sample Period $T_{s,j}$ | Time Step (s) |
| Number of Samples $l_j$ | Time Horizon (s) |
| Final Iteration $i_j^*$ | Discounting $\gamma$ |
| Ref Cmd $r_j$ | Network Dimensions |
| Initial Controller $K_{0,j}$ | # Ensemble |
| | Activation |
| | Learning Rate |
| | Weight Decay |
| | Hidden Layer Gain |
| | Output Layer Gain |
| | Output Layer Bias |
| | Diagonal Softplus Gain $\beta_L$ |
| | Batch Size |
| | # Batches |
| | Eligibility Trace |
| | $n$-step Trace Weight |
| | # Iterations |
| | # Epochs/Iteration |
| | State Adversary $\|\xi_x\|_{\max}$ |
| | Action Adversary $\|\xi_u\|_{\max}$ |
| | Model Adversary $\|\xi_\theta\|_{\max}$ |
| | Obs Adversary $\|\xi_o\|_{\max}$ |

designers. Firstly, RCI requires less than 5 s to run regardless of the system, as opposed to FVI's 6,000-9,000 s. This drastically shortened training time allows designers to immediately iterate on hyperparameter selections. When combined with the loop-level selection capability discussed in Remark C.5, RCI offers far superior troubleshooting and transparency of design.

Table 9: RCI vs FVI: Algorithm time/data complexity

| Parameter | Pendulum | | Jet Aircraft | | DDMR | |
|---|---|---|---|---|---|---|
| | RCI | cFVI/rFVI | RCI | cFVI/rFVI | RCI | cFVI/rFVI |
| # Simulations Req'd | 1 | 1.05e+7 / 3.84e+6 | 1 | 5.12e+6 | 1 | 5.12e+6 |
| # Data Samples Req'd | 15 | 3.45e+8 / 1.73e+8 | 45 | 2.30e+8 | 35 | 2.30e+8 |
| # NN Weights | 3 | 79,104 | 13 | 79,104 | 6 | 79,104 |
| # Epochs Req'd | 5 | 2,000 / 3,000 | 5 | 2,000 | 5 | 2,000 |
| Avg Training Time (s) | 0.17[a] | 6.88e+3[b] / 8.98e+3 | 4.25 | 8.18e+3 / 7.98e+3 | 2.58 | 6.30e+3 / 6.04e+3 |
| # Trials/Seeds Tested | 1,620[c] | 20 | 1,089 | 20 | 1,287 | 20 |

[a]Averaged over initial condition sweep $x_0 \in G_{x_0}$ on the nominal model $\nu = 1$.
[b]Averaged over 20 seeds (cf. Appendix H).
[c]RCI sweeps over the IC grid $x_0 \in G_{x_0}$ and modeling error grid $\nu \in G_\nu$ (see Section 3.2).

One number is larger for RCI than the FVI algorithms: the number of learning trials conducted. Since RCI can be run so quickly, for the ablation studies in Section 4 we have the luxury of running RCI at almost 160 times the number of learning trials than cFVI and rFVI. This should stand as a clear empirical benchmarking advantage of classical methods.

**Remark C.7 (On RCI vs FVI Evaluation Performance: Key Insights)** We conclude this section with a summary of the key takeaways from the numerical analysis presented in Sections 4, 5. These studies crucial new insights into algorithm structure's impact on learning performance, with significant practical takeaways for real-world designers. Kleinman's quadratic basis keeps RCI's

network dimension low and enables its theoretical guarantees, but consequently FVI enjoys an approximation advantage where system nonlinearities are strong. Regardless, RCI exhibits a decisive performance advantage for the higher-order, multi-loop systems studied in this work, suggesting that: 1) Our physics-based decentralization paradigm developed for these systems trumps FVI's approximation advantage, and 2) FVI's large data learning struggles as system order/complexity is increased. Indeed, FVIs require three orders of magnitude more time and six orders of magnitude more simulations to train than RCI, suggesting these algorithms face the common deep RL time/data complexity challenges. As a direct result, the deep methods require training in computer simulation on a nominal model, which makes cFVI particularly sensitive to modeling error performance degradation. rFVI attempts to robustify this vulnerability by training under adversarial input (Lutter et al., 2022), which ultimately mitigates degradation at the cost of inferior overall performance. By contrast, RCI's classical formulation reduces data requirements to a single simulation trial collected from the actual model, enabling more focused policy training.

These insights motivate an important practical consideration for designers: Given the uncertain nature of modeling error, it is perhaps unclear *a priori* which FVI method is appropriate for a given application. As a result, designers must make a cumbersome decision between two tools without clear criteria for their use cases, and they face empirically-demonstrated repercussions even if they make the right decision. By contrast, classical principles furnish a single RCI framework offering designers both excellent closed-loop performance (via its theoretical guarantees) and little degradation (via its focused training and inherited LQ robustness characteristics (Rodriguez, 2004)).

## APPENDIX D    PENDULUM MODEL & DESIGN FRAMEWORK

### D.1    PENDULUM MODEL

We consider the identical pendulum model used in the cFVI evaluations (Lutter et al., 2023a; 2022) for this work, which has the following equations of motion

$$\dot{\theta} = \omega,$$
$$\dot{\omega} = \frac{mgL}{2I} \sin\theta + \frac{1}{I}\tau, \tag{21}$$

where $\theta$ is the pendulum angle (measured zero pointing upward, positive counterclockwise), $\omega$ is the pendulum angular velocity, and $\tau$ is the torque applied to the pendulum base. The numerical values of all model constants are chosen identical to the cFVI evaluations (Lutter et al., 2023a; 2022) and are available in Table 10.

The system (21) is second-order, with states $x = [\theta, \omega]^T$ and control $u = \tau$. We examine the output $y = \theta$; i.e., control of the pendulum angle $\theta$. We examine the upright pendulum equilibrium $x_e = [\theta_e, \omega_e]^T = [0 \text{ rad}, 0 \text{ rad/s}]^T$. At this upright condition, the pendulum is trimmed by the control $\tau_e = 0$ N-m.

Table 10: Pendulum model parameters

| Definition | Symbol | Value |
|---|---|---|
| Pendulum length | $L$ | $L_0 = 1$ m (nominal) |
| Pendulum mass | $m$ | 1 kg |
| Gravitational field constant | $g$ | 9.81 m/s$^2$ |
| Pendulum moment of inertia | $I$ | $\frac{1}{3}mL^2$ |

**Remark D.1 (Pendulum Dynamical Structure)** The pendulum length $L$ is a central physical parameter in the dynamics (21). Firstly, increasing the pendulum length $L$ increases its rotational inertia $I$ in the square of the length; resultantly, (21) shows that the torque $\tau$ required to achieve the same angular acceleration increases with the square of the pendulum length $L$. The pendulum length $L$ also determines the severity of the upright pendulum instability. Linearization of (21) about the upright equilibrium yields

$$\begin{bmatrix} \dot{\theta} \\ \dot{\omega} \end{bmatrix} = \begin{bmatrix} 0 & 1 \\ \frac{mgL}{2I} & 0 \end{bmatrix} \begin{bmatrix} \theta \\ \omega \end{bmatrix} + \begin{bmatrix} 0 \\ \frac{1}{I} \end{bmatrix} \tau. \tag{22}$$

Examination the linearization (22) shows that the system has modes

$$s = \pm\sqrt{\frac{3}{2}}\sqrt{\frac{g}{L}}. \tag{23}$$

The real, imaginary-axis-symmetric pair of poles (23) is a common feature of inverted pendulum systems. We notice that the bandwidth of these modes are inversely proportional to the square root of the pendulum length $L$; i.e., a shorter pendulum increases the instability and system bandwidth. A longer pendulum reduces the instability, but it also reduces system bandwidth. Combined with the above discussion of the reduced control effectiveness associated with increased pendulum length, these first-principles analyses have significant implications for practical robotics design. Generally speaking, taller robotic systems with inverted pendulum instabilities will require significantly more actuator effort to achieve control objectives than shorter systems.

In the studies conducted in this work, we will focus on how modeling errors in the pendulum length $L$ affect the pendulum dynamics and learning performance. Specifically, we study modeling errors of the form

$$L = \nu\, L_0, \tag{24}$$

where $L_0 \in \mathbb{R}$ is a nominal value of the pendulum length, and $\nu \in \mathbb{R}$ is the modeling error parameter (nominally 1). As $\nu > 1$ increases, $L > L_0$ increases, and our prior discussion shows that the system becomes more sluggish and requires greater control effort.

Table 11 shows the inverted pendulum instability and control effectiveness constant $\frac{1}{I}$ as a function of the modeling error $\nu$ (24). As predicted in (23), the system instability reduces with increasing pendulum length. Control effectiveness is highly sensitive to changes in pendulum length, decreasing by 17% for a 10% modeling error $\nu = 1.1$ and by 36% for a 25% modeling error $\nu = 1.25$. Increases in the pendulum length will thus result in degraded closed-loop performance.

Table 11: Pendulum instability and control effectiveness versus modeling error parameter $\nu$ (24)

| $\nu$ (24) | Unstable Mode Location (23) | Control Effectiveness $\frac{1}{I}$ |
|---|---|---|
| 1 (nom) | 3.8360 | 3 |
| 1.1 | 3.6575 | 2.4793 |
| 1.25 | 3.4310 | 1.9200 |

### D.2 PENDULUM DESIGN FRAMEWORK

The pendulum system (21) is fundamentally a single-loop system $j = 1$. Thus, we do not employ the multi-loop decentralization techniques of RCI for this system. We do wish to highlight the great dynamical flexibility of RCI discussed in Section 2: RCI generalizes to any integer number of loops $j \in \mathbb{N}$, and this includes the single-loop case $j = 1$. The optimal LQ controller $K_1^*$ is a function of the modeling error $\nu$, so when necessary we will show explicit dependence by the notation $K_1^*(\nu)$. For the model parameters in Table 10 and cost structure selections (58), the pendulum has the following optimal LQ controllers

$$K_1^*(1) = \begin{bmatrix} 10.0098 & 2.6217 \end{bmatrix}, \tag{25}$$

$$K_1^*(1.1) = \begin{bmatrix} 10.9733 & 3.0086 \end{bmatrix}, \tag{26}$$

$$K_1^*(1.25) = \begin{bmatrix} 12.4235 & 3.6251 \end{bmatrix}, \tag{27}$$

As can be seen, the optimal controller $K_1^*(\nu)$ is heavily dependent on the modeling error $\nu$ (24).

## APPENDIX E    JET AIRCRAFT MODEL & DESIGN FRAMEWORK

### E.1    JET AIRCRAFT MODEL

Consider the following T-tailed small jet airplane model (Stengel, 2022; Soderman & Aiken, 1971)

$$
\begin{bmatrix} \dot{V} \\ \dot{\gamma} \\ \dot{q} \\ \dot{\alpha} \end{bmatrix} = \begin{bmatrix} -D_V & -g\cos\alpha_e & 0 & 0 \\ \frac{L_V}{V_e} & 0 & 0 & \frac{L_\alpha}{V_e} \\ 0 & 0 & M_q & M_\alpha \\ \frac{-L_V}{V_e} & 0 & 1 & \frac{-L_\alpha}{V_e} \end{bmatrix} \begin{bmatrix} V \\ \gamma \\ q \\ \alpha \end{bmatrix} + \begin{bmatrix} T_{\delta_T} & 0 \\ 0 & 0 \\ 0 & M_{\delta_E} \\ 0 & 0 \end{bmatrix} \begin{bmatrix} \delta_T \\ \delta_E \end{bmatrix}, \quad (28)
$$

where $V$ is the vehicle airspeed, $\gamma$ is the flightpath angle (FPA), $q$ is the pitch rate, and $\alpha$ is the vehicle angle of attack (AOA). As is standard in aerospace circles, here a subscript denotes a partial derivative with respect to the particular variable (e.g., $D_V$ denotes the dimensional aerodynamic derivative of drag $D$ with respect to airspeed $V$). For definitions of the parameters and their numerical values, see Table 12. This jet airplane model is a central example of the standard flight control text Stengel (2022), and it was constructed from aerodynamic data obtained by full-scale wind tunnel tests conducted by NASA (Soderman & Aiken, 1971).

The jet (28) is fourth-order, with states $x = [V, \gamma, q, \alpha]^T$ and controls $u = [\delta_T, \delta_E]^T$. We examine a level steady flight condition $\gamma_e = 0, q_e = 0$ at a cruising airspeed $V_e = 100$ m/s and altitude $h_e = 1000$ m (Mach $M_e \approx 0.3$). At this flight condition, the vehicle is trimmed at an angle of attack $\alpha_e = 3.4006$ deg by the controls $u_e = [\delta_{T,e}, \delta_{E,e}]^T = [0.2135, 0 \text{ deg}]^T$.

Table 12: Jet aircraft model parameters

| Definition | Symbol | Value |
|:---:|:---:|:---:|
| Lift/AOA aero deriv | $L_\alpha$ | $L_{\alpha 0} = 127.9$ N/rad (nominal) |
| Lift/airspeed aero deriv | $L_V$ | 0.190 N/(m/s) |
| Drag/airspeed aero deriv | $D_V$ | 1.850 N/(m/s) |
| Moment/AOA aero deriv | $M_\alpha$ | -798.56 N-m/rad |
| Moment/pitch rate aero deriv | $M_q$ | -127.94 N-m/(rad/s) |
| Thrust/throttle setting control deriv | $T_{\delta_T}$ | 4.6645 N/- |
| Moment/elevator control deriv | $M_{\delta_E}$ | -9.069 N-m/rad |
| Gravitational field constant | $g$ | 9.81 m/s$^2$ |

**Remark E.1 (Jet Aircraft Minimum Phase Behavior)** We note in (28) that the $(2,2)$ element of the input gain matrix $B$ is assumed zero; i.e., elevator deflections $\delta_E$ do not directly impact the FPA derivative $\dot{\gamma}$. As a result, the jet model (28) is minimum phase. However, in reality tail-controlled aircraft feature lift/elevator parasitic couplings in this location which cause them to be nonminimum phase (Rodriguez et al., 2008; Bolender & Doman, 2006). Nevertheless, the assumption made in the development of this model (Stengel, 2022) is quite standard in modeling for flight control design (Hauser et al., 1992; Marrison & Stengel, 1998; Wang & Stengel, 2000).

**Remark E.2 (Jet Aircraft Decentralized Dynamical Structure)** The jet aircraft studied here is a multi-input system which naturally lends itself to a decentralized dynamical structure. The throttle $\delta_T$ is associated with the airspeed $V$ in the translational loop $j = 1$, and the elevator $\delta_E$ is associated with the FPA $\gamma$ and attitude $q, \alpha$ in the rotational loop $j = 2$. Indeed, this decentralized structure is general to aviation systems (Stengel, 2022), even high-performance hypersonic vehicles (HSVs) (Dickeson et al., 2009a;b; Dickeson, 2012; Marrison & Stengel, 1998; Wang & Stengel, 2000; Parker et al., 2006).

The lift/AOA derivative $L_\alpha$ determines the lift efficiency of the aircraft and is hence a central aerodynamic parameter in any aviation system. As with any aerodynamic modeling process, it is also subject to large modeling errors and sensitivity to changes in flight condition (Stengel, 2022; Bolender & Doman, 2007). Thus, in this work we study the effects of modeling error $\nu$ on the lift/AOA dimensional aerodynamic derivative as

$$
L_\alpha = \nu L_{\alpha 0}, \quad (29)
$$

where $L_{\alpha 0} \in \mathbb{R}$ is the nominal value of the lift/AOA aerodynamic derivative, and $\nu \in \mathbb{R}$ is the modeling error parameter (nominally 1). As $\nu < 1$ decreases, the vehicle exhibits decreased lift

efficiency, leading to a more difficult control problem (Stengel, 2022). The jet aircraft (28) has the following characteristic equation and natural modes

$$\phi(s) = \left(s^2 + 2\zeta_{ph}s + \omega_{n_{ph}}\right)\left(s^2 + 2\zeta_{sp}s + \omega_{n_{sp}}\right), \qquad (30)$$

$$s_{ph} = -\zeta_{ph}\omega_{n_{ph}} \pm j\omega_{n_{ph}}\sqrt{1 - \zeta_{ph}^2}, \qquad s_{sp} = -\zeta_{sp}\omega_{n_{sp}} \pm j\omega_{n_{sp}}\sqrt{1 - \zeta_{sp}^2}. \qquad (31)$$

The first pair of modes $s_{ph}$ (31) is the phugoid mode, associated with the translational dynamics via exchanges in kinetic and potential energy (i.e., with coupled oscillations between the airspeed $V$ and FPA $\gamma$). They are generally slow and lightly damped. The second pair $s_{sp}$ (31) is called the short-period mode and is associated with the rotational dynamics via exchanges between rotational energy and aeroelastic energy (i.e., with coupled oscillations between the pitch rate $q$ and AOA $\alpha$). If the vehicle is designed so the center of gravity (c.g.) lies forward the center of pressure (c.p.), they are generally fast, stable, and lightly damped (as is the case here). If the c.p. lies forward the c.g., they are generally real, imaginary-axis symmetric, one stable and the other unstable.

We have plotted these modes as a function of the modeling error paramer $\nu$ (29) in Table 13. As can be seen, the damping of both modes decreases with decreased lift efficiency $\nu < 1$, and the short-period modes get closer to the imaginary axis; i.e., less stable.

Table 13: Jet aircraft phugoid and short-period modes versus modeling error parameter $\nu$ (29)

| $\nu$ (29) | $s_{ph}$ | $\zeta_{ph}$ | $\omega_{n_{ph}}$ | $s_{sp}$ | $\zeta_{sp}$ | $\omega_{n_{sp}}$ |
|---|---|---|---|---|---|---|
| 1 (nom) | $-0.00849 \pm j0.119$ | 0.0709 | 0.120 | $-1.28 \pm j2.83$ | 0.412 | 3.10 |
| 0.9 | $-0.00853 \pm j0.120$ | 0.0708 | 0.121 | $-1.22 \pm j2.83$ | 0.395 | 3.08 |
| 0.75 | $-0.00863 \pm j0.120$ | 0.0705 | 0.122 | $-1.12 \pm j2.82$ | 0.369 | 3.04 |

### E.2 JET AIRCRAFT DECENTRALIZED DESIGN FRAMEWORK

This work implements a decentralized design methodology, wherein controllers are designed separately for the weakly-coupled translational subsystem (associated with the airspeed $V$ and throttle setting $\delta_T$) and rotational subsystem (associated with the FPA $\gamma$, attitude $q, \alpha$, and elevator $\delta_E$). In order to achieve zero steady-state error to step reference commands, we augment the plant at the output with the integrator bank $z = \int y\, d\tau = [z_V, z_\gamma]^T = \left[\int V\, d\tau, \int \gamma\, d\tau\right]^T$. For RCI, the state/control vectors are thus partitioned as $x_1 = [z_V, V]^T$, $u_1 = \delta_T$ ($n_1 = 2$, $m_1 = 1$) and $x_2 = [z_\gamma, \gamma, q, \alpha]^T$, $u_2 = \delta_E$ ($n_2 = 4$, $m_2 = 1$). Applying the LQ servo design framework (Rodriguez, 2004) to each of the loops yields a proportional-integral (PI) speed controller $K_1^*$ and a PI/PD FPA/attitude inner-outer loop controller $K_2^*$. It is these optimal LQ controller parameters which RCI will learn online. In general, the optimal LQ controllers $K_j^*$ ($j = 1, 2$) are functions of the modeling error $\nu$, so when necessary we will show explicit dependence by the notation $K_j^*(\nu)$. For the model parameters in Table 12 and cost structure selections (59), the jet aircraft has the following optimal LQ controllers

$$K_1^*(\nu) = [\ 0.0316 \quad 0.1496\ ], \qquad (32)$$

$$K_2^*(1) = [\ -0.7071 \quad -1.7089 \quad -0.1960 \quad -0.4251\ ], \qquad (33)$$

$$K_2^*(0.9) = [\ -0.7071 \quad -1.7403 \quad -0.1858 \quad -0.3944\ ], \qquad (34)$$

$$K_2^*(0.75) = [\ -0.7071 \quad -1.8014 \quad -0.1687 \quad -0.3450\ ]. \qquad (35)$$

As can be seen from (32), the optimal controller $K_1^*$ in the translational loop $j = 1$ is independent of the modeling error $\nu$. This is because the lift/AOA derivative $L_\alpha$ enters dynamically into the FPA $\gamma$ and AOA $\alpha$ equations in (28), so the modeling error only affects the dynamics in the rotational loop $j = 2$.

## APPENDIX F DIFFERENTIAL DRIVE MOBILE ROBOT (DDMR) MODEL & DESIGN FRAMEWORK

### F.1 DDMR MODEL

Consider the following DDMR model (Dhaouadi & Abu Hatab, 2013; Mondal et al., 2020; 2019; Mondal, 2021)

$$
\begin{aligned}
\dot{V} &= \frac{-2\overline{\beta}}{\hat{m}r^2}V + \frac{m_c d}{\hat{m}}\omega^2 + \frac{k_t}{\hat{m}k_g r}i_{a_r} + \frac{k_t}{\hat{m}k_g r}i_{a_l}, \\
\dot{\omega} &= -\frac{\overline{\beta}d_w^2}{2\hat{I}r^2}\omega - \frac{m_c d}{\hat{I}}\omega V + \frac{d_w k_t}{2\hat{I}k_g r}i_{a_r} - \frac{d_w k_t}{2\hat{I}k_g r}i_{a_l} \\
\dot{i}_{a_r} &= \frac{-k_g k_b}{l_a r}V - \frac{k_g k_b d_w}{2l_a r}\omega - \frac{r_a}{l_a}i_{a_r} + \frac{1}{2l_a}\overline{e}_a + \frac{1}{2l_a}\Delta e_a \\
\dot{i}_{a_l} &= \frac{-k_g k_b}{l_a r}V + \frac{k_g k_b d_w}{2l_a r}\omega - \frac{r_a}{l_a}i_{a_l} + \frac{1}{2l_a}\overline{e}_a - \frac{1}{2l_a}\Delta e_a
\end{aligned}
\tag{36}
$$

where $V$ is the robot speed (measured positive forward), $\omega$ is the robot angular velocity (measured positive counterclockwise), and $i_{a_r}$, $i_{a_l}$ are the right and left DC motor armature currents, respectively. We provide definitions and numerical values of all model constants in Table 14. It should be noted that these parameter selections are standard and were obtained empirically from actual hardware (Mondal, 2021).

The system (36) is fourth-order, with states $x = [V, \omega, i_{a_r}, i_{a_l}]^T$. The controls are $u = [\overline{e}_a, \Delta e_a]^T$, where $\overline{e}_a = \frac{e_{a,r} + e_{a,l}}{2}$ is the average of the armature voltages $e_{a,r}, e_{a,l}$ applied to the right and left wheels, respectively, and $\Delta e_a = e_{a,r} - e_{a,l}$ is the difference of the armature voltages. We examine the outputs $y = [V, \omega]^T$; i.e., control of the DDMR speed $V$ and angular velocity $\omega$. We examine the equilibrium forward cruise condition $x_e = [V_e, \omega_e, i_{a_r,e}, i_{a_l,e}]^T = [2 \text{ m/s}, 0 \text{ rad/s}, 0.68\text{A}, 0.68\text{A}]^T$. At this cruise condition, the DDMR is trimmed by the controls $\overline{e}_{a,e} = 3.9115 \text{ V}, \Delta e_{a,e} = 0 \text{ V}$.

Table 14: DDMR model parameters

| Definition | Symbol | Value |
|---|---|---|
| c.g./wheelbase separation | $d$ | $d_0 = $ -6 cm (nominal) |
| Mass of robot chassis | $m_c$ | 3.963 kg |
| Mass of single wheel | $m_w$ | 0.659 kg |
| Wheel motor moment of inertia | $I_w$ | 570 $\mu$kg-m$^2$ |
| Total vehicle moment of inertia | $I$ | 0.224 kg-m$^2$ |
| Radius of wheels | $r$ | 3.85 cm |
| Length of robot chassis | $l$ | 44 cm |
| Width of robot chassis | $w$ | 34 cm |
| Distance between wheels at midpoint | $d_w$ | 34 cm |
| Motor armature inductance | $l_a$ | 13.2 $\mu$H |
| Motor armature resistance | $r_a$ | 3.01 Ohm |
| Motor gear up/down ratio | $k_g$ | 1 |
| Motor back EMF constant | $k_b$ | 0.075 V/(rad/s) |
| Motor torque constant | $k_t$ | 0.075 (N-m)/A |
| Speed damping constant | $\beta$ | 7.4 $\mu$N-m-s |
| Total vehicle mass | $m$ | $m_c + 2m_w$ |
| Effective mass | $\hat{m}$ | $m + \frac{2I_w}{r^2}$ |
| Effective moment of inertia | $\hat{I}$ | $I + \frac{d_w^2 I_w}{2r^2}$ |
| Effective damping constant | $\overline{\beta}$ | $\beta + \frac{k_t k_b}{r_a}$ |

**Remark F.1 (DDMR Dynamical Structure)** Assuming that the motor armature inductance $l_a \approx 0$ (which has proven a reasonable approximation – if included, the motor dynamics have poles on the

order of $s = -10^6$), then the DDMR model (36) reduces to (Mondal, 2021)

$$\dot{V} = \frac{-2\overline{\beta}}{\hat{m}r^2}V + \frac{m_c d}{\hat{m}}\omega^2 + \frac{2k_t}{\hat{m}k_g r_a r}\overline{e}_a,$$

$$\dot{\omega} = -\frac{\overline{\beta}d_w^2}{2\hat{I}r^2}\omega - \frac{m_c d}{\hat{I}}\omega V + \frac{d_w k_t}{2\hat{I}k_g r_a r}\Delta e_a. \tag{37}$$

We will use this model for purposes of first-principles analysis here. We also use it as the design model for the methods studied to improve numerics. Examination of the DDMR model (37) quickly reveals a natural dynamical partition of the form (14). The translational loop $j = 1$ consists of the speed state $V$ and is associated with the average voltage control $\overline{e}_a$. The rotational loop $j = 2$ consists of the angular velocity state $\omega$ and is associated with the differential voltage control $\Delta e_a$.

The (signed) distance $d$ that the vehicle center of gravity (c.g.) lies forward the wheelbase is a central physical parameter in the dynamics of the DDMR (37). Firstly, the c.g./wheelbase separation $d$ determines the strength of the coupling terms in (37) (i.e., the second term in each state equation). Indeed, (37) shows that when $d = 0$, the translational and rotational dynamics of the DDMR decouple – why placing the robot c.g. on the wheel axis is a common design choice in the DDMR community (Mondal et al., 2020; 2019; Mondal, 2021).

The c.g./wheelbase separation $d$ also determines the stability properties of the DDMR. Loosely speaking, placing the vehicle c.g. forward the wheelbase $d >> 0$ renders the rotational dynamics $\omega$ stable: Perturbations in the robot's rotational pose make the friction forces acting on the wheelbase induce torques on the vehicle which counter the direction of the perturbation. Conversely, placing the wheelbase forward the c.g. $d << 0$ results in directional instability for similar reasons. This stability behavior is entirely analogous to the longitudinal dynamics of aircraft, wherein pitch-up instabilities occur if and only if the vehicle center of pressure (playing the analogous role of the wheelbase as the center of forces acting on the vehicle) lies forward the c.g. (Stengel, 2022) (see Appendix E).

More concretely, given an equilibrium $x_e = [V_e, \omega_e]^T$ of (37), the following controls $u_e = [\overline{e}_{a,e}, \Delta e_{a,e}]^T$ achieve equilibrium

$$\overline{e}_{a,e} = -\frac{\hat{m}k_g r_a r}{2k_t}\left(-\frac{2\overline{\beta}}{\hat{m}r^2}V_e + \frac{m_c d}{\hat{m}}\omega_e^2\right),$$

$$\Delta e_{a,e} = \frac{2\hat{I}k_g r_a r}{d_w k_t}\left(\frac{m_c d}{\hat{I}}V_e\omega_e + \frac{\overline{\beta}d_w^2}{2\hat{I}r^2}\omega_e\right), \tag{38}$$

and linearization about the equilibrium $(x_e, u_e)$ yields

$$\begin{bmatrix} \dot{V} \\ \dot{\omega} \end{bmatrix} = \begin{bmatrix} \frac{-2\overline{\beta}}{\hat{m}r^2} & \frac{2m_c d\omega_e}{\hat{m}} \\ \frac{-m_c d\omega_e}{\hat{I}} & \left(\frac{-m_c dV_e}{\hat{I}} - \frac{\overline{\beta}d_w^2}{2\hat{I}r^2}\right) \end{bmatrix} \begin{bmatrix} V \\ \omega \end{bmatrix} + \begin{bmatrix} \frac{2k_t}{\hat{m}k_g r_a r} & 0 \\ 0 & \frac{d_w k_t}{2\hat{I}k_g r_a r} \end{bmatrix} \begin{bmatrix} \overline{e}_a \\ \Delta e_a \end{bmatrix}. \tag{39}$$

Note from examination of the linearization (39) that, reaffirming our insights of the nonlinear dynamics (37), the DDMR decouples when $d = 0$; i.e., when the vehicle c.g. is placed on the wheelbase. Decoupling of the *linearized* model also occurs when $\omega_e = 0$ (studied here), in which case examination (39) shows that the DDMR has open-loop eigenvalues at $s = s_V, s_\omega$, where

$$s_V \triangleq \frac{-2\overline{\beta}}{\hat{m}r^2}, \tag{40}$$

$$s_\omega \triangleq -\left(\frac{m_c V_e}{\hat{I}}d + \frac{\overline{\beta}d_w^2}{2\hat{I}r^2}\right). \tag{41}$$

The first mode $s_V$ (40) is a stable speed damping mode arising from wheel friction and the electro-mechanical damping characteristics of the motors. The second mode $s_\omega$ (41) determines the stability properties of the rotational dynamics. (41) shows that the following critical value $d_{MS}$ of the c.g./wheelbase separation results in marginal stability:

$$d_{MS} = -\frac{\overline{\beta}d_w^2}{2m_c V_e r^2}. \tag{42}$$

For $d > d_{MS}$, the DDMR is stable. For $d < d_{MS}$, the DDMR is unstable. Note in the case of zero speed/motor damping $\overline{\beta} = 0$ that $d_{MS} = 0$; i.e., the DDMR is stable if and only the vehicle c.g. lies forward the wheel axis $d > 0$ – numerically reaffirming the physical intuitions discussed above. More generally, (41) shows that greater translational damping $\overline{\beta}$ increases the stability of the DDMR rotational dynamics. For the DDMR parameters studied (cf. Table 14), $d_{MS} = -0.92$ cm, so the nominal c.g./wheelbase separation $d_0 = -6$ cm results in a directionally-unstable system.

In the studies conducted in this work, we will focus on how modeling errors in the c.g./wheelbase separation $d$ affect the DDMR dynamics and learning performance. Specifically, we study modeling errors of the form

$$d = \nu \, d_0, \tag{43}$$

where $d_0 \in \mathbb{R}$ is a nominal value of the c.g./wheelbase separation, and $\nu \in \mathbb{R}$ is the modeling error parameter (nominally 1). As $\nu > 1$ increases, $d < d_0 < d_{MS} < 0$ decreases, and (41) shows that the system becomes more unstable. Table 15 shows the effect of DDMR eigenvalues as a function of the modeling error parameter $\nu$ (43). The DDMR directional instability is highly sensitive to modeling errors $\nu$ in the c.g./wheelbase separation $d$. As predicted by (40), the speed mode $s_V$ is stable and independent of the c.g/wheelbase separation $d$.

Table 15: DDMR eigenvalues versus modeling error parameter $\nu$ (43)

| $\nu$ (43) | Speed Mode $s_V$ (40) | Angular Velocity Mode $s_\omega$ (41) |
|---|---|---|
| 1 (nom) | -0.4184 | 1.6343 |
| 1.1 | -0.4184 | 1.8274 |
| 1.25 | -0.4184 | 2.1171 |

### F.2 DDMR DECENTRALIZED DESIGN FRAMEWORK

This work implements a decentralized design methodology, wherein controllers are designed separately for the weakly-coupled translational subsystem (associated with the speed $V$ and average voltage control $\overline{e}_a$) and rotational subsystem (associated with the angular velocity $\omega$ and differential voltage control $\Delta e_a$). In order to achieve zero steady-state error to step reference commands, we augment the plant at the output with the integrator bank $z = \int y \, d\tau = [z_V, \, z_\omega]^T = \left[\int V \, d\tau, \, \int \omega \, d\tau\right]^T$. For RCI, the state/control vectors are thus partitioned as $x_1 = [z_V, \, V]^T$, $u_1 = \overline{e}_a$ ($n_1 = 2$, $m_1 = 1$) and $x_2 = [z_\omega, \, \omega]^T$, $u_2 = \Delta e_a$ ($n_2 = 2$, $m_2 = 1$). Applying the LQ servo design framework (Rodriguez, 2004) to each of the loops yields a proportional-integral (PI) speed controller $K_1^*$ and a PI angular velocity controller $K_2^*$. It is these optimal LQ controller parameters which RCI will learn online. For the model parameters in Table 14 and cost structure selections (60), the DDMR has the following optimal LQ controllers

$$K_1^*(\nu) = [\ 3.6515 \quad 5.2062\ ], \tag{44}$$

$$K_2^*(1) = [\ 5.0000 \quad 10.2344\ ], \tag{45}$$

$$K_2^*(1.1) = [\ 5.0000 \quad 10.9164\ ], \tag{46}$$

$$K_2^*(1.25) = [\ 5.0000 \quad 11.9718\ ]. \tag{47}$$

As can be seen from (44), the optimal controller $K_1^*$ in the translational loop $j = 1$ is independent of the modeling error $\nu$. This is immediately seen from examination of the linearized dynamics (39), wherein we observe that the diagonal terms in $A$, $B$ pertaining to the speed $V$ are independent of the c.g./wheelbase separation $d$, hence of the modeling error $\nu$. On the other hand, the optimal controller $K_2^*$ in the rotational loop $j = 2$ is heavily dependent on the modeling error $\nu$.

## APPENDIX G  EXPLORATION STUDIES SETUP, HYPERPARAMETER SELECTIONS

**Hardware.** These studies were performed in MATLAB R2022b, on an NVIDIA RTX 2060, Intel i7 (9th Gen) processor. All numerical integrations in this work are performed in MATLAB's adaptive `ode45` solver to ensure solution accuracy.

**Software.** All RCI code developed for this work is available at Anonymized (2024). All FVI results (Lutter et al., 2023a; 2022) were generated from the open-source repository developed by the authors Lutter et al. (2023b).

**Training Procedures.** Throughout the comparative CT-RL studies presented in Section 5, we present data for RCI trained at the equilibrium initial condition $x_0 = x_e$ for the respective modeling error $\nu = 1, 1.1, 1.25$. Since RCI requires only one simulation (cf. Table 9) and on the order of $l = 20$ data points (cf. Table 16) to train to completion, online learning is highly practical for this algorithm. On the other hand, FVI requires on the order of 5 million simulations to achieve good learning performance (cf. Table 9 and original studies (Lutter et al., 2023a; 2022)). Nowhere near the required 5 million trials needed to properly learn the modeling error can be executed in a real-world setting. As a result, the only practical means of training FVI is in simulation. Since the modeling error $\nu$ for a given system is not known *a priori*, this means that FVI must train on the nominal model (modulo adversary perturbations in the rFVI case (Lutter et al., 2022)). Thus, the results presented in this work for FVI are attained through training on the nominal model $\nu = 1$, a training procedure identical to that presented by the original authors in (Lutter et al., 2023a; 2022).

### G.1 DATA GENERATION, PERFORMANCE METRICS CONSIDERED

In the pendulum ablation study, we examine the convergence and conditioning performance of RCI with respect to pendulum initial conditions $x_0 = [\theta_0, \, \omega_0]^T$ in the grid

$$G_{x_0} = [-60 : 15 : 60] \text{ deg } \times [-60 : 15 : 60] \text{ deg/s.} \tag{48}$$

Note that, as is the case with all systems in this work, the pendulum IC grid $G_{x_0}$ is centered about studied equilibrium point $x_e$; namely, the upright equilibrium point $x_e = [\theta_e, \, \omega_e]^T = [0 \text{ deg}, \, 0 \text{ deg/s}]^T$ (cf. Appendix D). In the jet aircraft ablation study, we examine initial conditions $x_0 = [V_0, \gamma_0, q_0 = 0, \alpha_0 = 0]^T$ in the grid

$$G_{x_0} = [90 : 2 : 110] \text{ m/s } \times [-2 : 0.5 : 2] \text{ deg.} \tag{49}$$

In the DDMR ablation study, we examine initial conditions $x_0 = [V_0, \, \omega_0]^T$ in the grid

$$G_{x_0} = [1.5 : 0.125 : 2.5] \text{ m/s } \times [-30 : 5 : 30] \text{ deg/s.} \tag{50}$$

In the jet aircraft and DDMR studies, we always initialize the integrator augmentation states $z_0 = 0$ (cf. Appendices E.2 and F.2). For the pendulum, we study modeling errors $\nu$ (24) in the grid

$$G_\nu = [1 : 0.01 : 1.25], \tag{51}$$

while for the jet aircraft, we study modeling errors $\nu$ (29) in the grid

$$G_\nu = [1 : -0.025 : 0.75], \tag{52}$$

and for the DDMR, we study modeling errors $\nu$ (43) in the grid

$$G_\nu = [1 : 0.025 : 1.25]. \tag{53}$$

In what follows, consider an arbitrary system, and let a loop $j \in \mathbb{N}$ be given. In loop $j$, we define the final-iteration controller error $e_{K_j}$ as

$$e_{K_j}(x_0, \nu) = \left\| K_{i^*, j}(x_0, \nu) - K_j^*(\nu) \right\|, \quad (x_0, \nu) \in G_{x_0} \times G_\nu, \tag{54}$$

where $K_{i^*, j}$ is the final-iterate RCI controller in loop $j$, $K_j^*$ is the optimal LQ controller in loop $j$, and the norm used in (54) and subsequently is the operator norm. The final RCI controller $K_{i^*, j}$ is, strictly speaking, a function of the initial condition $x_0$ (which determines the state-action data available for learning) and modeling error $\nu$ (which determines the dynamics and hence also the state trajectory produced by the closed-loop system). Meanwhile, the optimal LQ controller $K_j^*$ is a function of the model parameters $\nu$ (43) only (omitting obvious dependence on the state/control penalties $Q_j$, $R_j$ for our purposes here).

Given a modeling error $\nu \in \mathbb{R}$, we define the nominal LQ controller error $e_{K_j, nom}$ as

$$e_{K_j, nom}(\nu) = \left\| K_j^*(1) - K_j^*(\nu) \right\|, \qquad \nu \in G_\nu. \tag{55}$$

This quantifies how severe the controller error is if the designer keeps the classical LQR design performed on the nominal model $\nu = 1$ without updating. Examining the size of this error relative to the final RCI controller error $e_{K_j}$ (54) is absolutely central and rigorously answers the question: How much better off is a designer by running RCI than by sticking with a nominal LQ design? To this end, we define the percentage controller error $\%e_{K_j}$ of RCI relative to the nominal as

$$\%e_{K_j}(x_0, \nu) = \left( \frac{e_{K_j}(x_0, \nu)}{e_{K_j, nom}(\nu)} \right) \times 100, \qquad (x_0, \nu) \in G_{x_0} \times G_\nu, \tag{56}$$

whenever this number is well-defined. Note that *smaller* percent controller error $\%e_{K_j}$ implies RCI performs *better* in comparison to the nominal classical design.

Studying the convergence performance of RCI for a fixed modeling error $\nu$, it is natural to define the IC-sweep worst-case final controller error $e_{K_j, \max}$ as

$$e_{K_j, \max}(\nu) = \max_{x_0 \in G_{x_0}} e_{K_j}(x_0, \nu), \qquad \nu \in G_\nu. \tag{57}$$

This metric provides an upper bound for the controller error expected from RCI for a fixed model uncertainty $\nu$ (43). It also has great practical utility: Given an *a priori* upper bound for the severity of the modeling error, $e_{K_j, \max}$ provides an upper bound for RCI's resulting final controller error.

## G.2 HYPERPARAMETER SELECTIONS

### G.2.1 SHARED HYPERPARAMETERS

**State, Control Penalty Gains.** For the pendulum, we use identical penalty selections to those in the original cFVI studies (Lutter et al., 2023a; 2022); namely,

$$Q_1 = \texttt{diag}(1, 0.1), \qquad\qquad R_1 = 0.5. \tag{58}$$

For the jet aircraft, consider the decentralized design framework described in Section E.2. We choose the following cost structure

$$\begin{aligned} Q_1 &= \texttt{diag}(0.005, 0.05), & R_1 &= 5, \\ Q_2 &= \texttt{diag}(0.5, 1, 0, 0), & R_2 &= 1. \end{aligned} \tag{59}$$

These state/control penalties were chosen to yield optimal LQ controllers $K_1^*$ (32), $K_2^*$ (33) achieving nominal closed-loop step response specifications comparable to existing benchmarks (Stengel, 2022): A 90% rise time in speed $t_{r,V,90\%} = 9.297$ s and FPA $t_{r,\gamma,90\%} = 4.52$ s, a 1% settling time in speed $t_{s,V,1\%} = 14.47$ s and FPA $t_{s,\gamma,1\%} = 7.20$ s, percent overshoot in speed $M_{p,V} = 0.09\%$ and FPA $M_{p,\gamma} = 0.25\%$.

For the DDMR, consider the decentralized design framework described in Section F.2. We choose the following cost structure

$$\begin{aligned} Q_1 &= 10 I_2, & R_1 &= 0.75, \\ Q_2 &= \texttt{diag}(25, 7.5), & R_2 &= 1. \end{aligned} \tag{60}$$

These state/control penalties were chosen to yield optimal LQ controllers $K_1^*$ (44), $K_2^*$ (45) achieving nominal closed-loop step response specifications comparable to existing benchmarks (Mondal et al., 2020; 2019): A 90% rise time in speed $t_{r,V,90\%} = 3.778$ s and angular velocity $t_{r,\omega,90\%} = 1.27$ s, a 1% settling time in speed $t_{s,V,1\%} = 5.556$ s and angular velocity $t_{s,\omega,1\%} = 6.73$ s, percent overshoot in speed $M_{p,V} = 0\%$ and angular velocity $M_{p,\omega} = 16.9\%$.

### G.2.2 RCI

**Initial Stabilizing Controller.** For the pendulum, we use the initial stabilizing controller

$$K_{0,1} = [\ 13.5108 \quad 5.8316\ ], \tag{61}$$

which we obtained from cost structure selections $Q_1 = \texttt{diag}(0.5, 0.25)$, and $R_1 = 0.01$. For the jet aircraft in loop $j$ ($j = 1, 2$), we use the initial stabilizing controllers

$$K_{0,1} = [\ 0.0316 \quad 0.1168\ ], \tag{62}$$

$$K_{0,2} = [\ -1.7321 \quad -3.4191 \quad -0.3427 \quad -0.9709\ ], \tag{63}$$

which we obtained from a decentralized design with cost structure selections $Q_1 = 0.01 I_2$, $R_1 = 10$, $Q_2 = \text{diag}(1.5, 2.5, 0, 0)$, and $R_2 = 0.5$. For the DDMR in loop $j$ ($j = 1, 2$), we use the initial stabilizing controllers

$$K_{0,1} = \begin{bmatrix} 2.2361 & 3.4966 \end{bmatrix}, \tag{64}$$

$$K_{0,2} = \begin{bmatrix} 8.6603 & 12.4403 \end{bmatrix}, \tag{65}$$

which we obtained from a decentralized design with cost structure selections $Q_1 = 5 I_2$, $R_1 = 1$, $Q_2 = \text{diag}(7.5, 2.5)$, and $R_2 = 0.1$.

The remainder of the RCI hyperparameter selections can be found in Table 16. Examination of Table 16 shows that these hyperparameter selections comprise little more than "round-number" designer first-choices, requiring only insights of the system dynamics and a few minutes of trial-and-error to obtain.

Table 16: RCI hyperparameter selections

| Hyperparameter | Pendulum | Jet Aircraft | | DDMR | |
| --- | --- | --- | --- | --- | --- |
| | Loop $j = 1$ | Loop $j = 1$ | Loop $j = 2$ | Loop $j = 1$ | Loop $j = 2$ |
| Sample Period $T_{s,j}$ (s) | 1 | 2 | 0.5 | 4 | 1 |
| Number of Samples $l_j$ | 15 | 15 | 30 | 20 | 15 |
| Final Iteration $i_j^*$ | 5 | 5 | 5 | 5 | 5 |
| Ref Cmd $r_j$ | $10\sin\left(\frac{2\pi}{10}t\right)$ | $5\sin\left(\frac{2\pi}{50}t\right)$ | $0.1\sin\left(\frac{2\pi}{2.5}t\right)$ | $2\sin\left(\frac{2\pi}{10}t\right)$ | $5\sin\left(\frac{2\pi}{50}t\right)$ |
| (deg $\mid$ m/s, deg $\mid$ m/s, deg/s) | $+5\sin\left(\frac{2\pi}{5}t\right)$ | $+10\sin\left(\frac{2\pi}{25}t\right)$ | $+0.1\sin\left(\frac{2\pi}{1.5}t\right)$ | $+\sin\left(\frac{2\pi}{5}t\right)$ | $+5\sin\left(\frac{2\pi}{5}t\right)$ |
| | | | | | $+5\sin\left(\frac{2\pi}{2.5}t\right)$ |
| Initial Controller $K_{0,j}$ | (61) | (62) | (63) | (64) | (65) |

### G.2.3    CFVI, RFVI

**Remark G.1 (cFVI/rFVI Hyperparameter Selections)**  As with our selections of the pendulum model structure and parameters (cf. Section D), for our pendulum studies we have selected hyperparameters identical to those of the original cFVI/rFVI evaluations (Lutter et al., 2023a; 2022), with two exceptions. In (Lutter et al., 2023a; 2022), the authors use a logcos control penalty function scaled so that its curvature at the origin $u = 0$ is $2R$; i.e., so that its curvature agrees with that of a quadratic penalty $u^T R u$. In order to make comparisons consistent across the methods studied, and in order to produce a more widely-applicable performance benchmark for real-world designers, we have decided to apply the standard quadratic control penalty $u^T R u$ for all methods. Likewise, the authors in (Lutter et al., 2023a; 2022) wrap the penalty function of the pendulum angle state to be periodic in $[0, 2\pi)$, a practice which we have dropped for consistency of comparison and generalizability of benchmarking. Finally, due to these changes we observed that more iterations were necessary for rFVI to converge in training the pendulum system (cf. Figure 6), so we increased its iteration count from 100 previously (Lutter et al., 2023a; 2022) to 150 here (cf. Table 17).

Hyperparameter selections for cFVI and rFVI can be found in Table 17. These parameter selections are overall quite standard and have indeed demonstrated great learning performance successes on second-order, unstable systems in previous studies (Lutter et al., 2023a; 2022).

Table 17: cFVI, rFVI hyperparameter selections

| Hyperparameter | Pendulum | | Jet Aircraft | | DDMR | |
|---|---|---|---|---|---|---|
| | cFVI | rFVI | cFVI | rFVI | cFVI | rFVI |
| Time Step (s) | 0.008 | 0.008 | 0.008 | 0.008 | 0.008 | 0.008 |
| Time Horizon (s) | 5 | 5 | 20 | 20 | 5 | 5 |
| Discounting $\gamma$ | 0.99 | 0.99 | 0.99 | 0.99 | 0.99 | 0.99 |
| Network Dimension | $[3 \times 96]$ | $[3 \times 96]$ | $[3 \times 96]$ | $[3 \times 96]$ | $[3 \times 96]$ | $[3 \times 96]$ |
| # Ensemble | 4 | 4 | 4 | 4 | 4 | 4 |
| Activation | Tanh | Tanh | Tanh | Tanh | Tanh | Tanh |
| Learning Rate | 1e-5 | 1e-5 | 3e-5 | 3e-5 | 3e-5 | 3e-5 |
| Weight Decay | 1e-6 | 1e-6 | 1e-6 | 1e-6 | 1e-6 | 1e-6 |
| Hidden Layer Gain | 1.41 | 1.41 | 1.41 | 1.41 | 1.41 | 1.41 |
| Output Layer Gain | 1.00 | 1.00 | 1.00 | 1.00 | 1.00 | 1.00 |
| Output Layer Bias | -0.1 | -0.1 | -0.1 | -0.1 | -0.1 | -0.1 |
| Diagonal Softplus Gain $\beta_L$ | 1.0 | 1.0 | 7.5 | 7.5 | 1.0 | 1.0 |
| Batch Size | 256 | 128 | 256 | 256 | 256 | 256 |
| # Batches | 200 | 200 | 200 | 200 | 200 | 200 |
| Eligibility Trace | 0.85 | 0.85 | 0.85 | 0.85 | 0.85 | 0.85 |
| $n$-step Trace Weight | 1e-4 | 1e-4 | 1e-4 | 1e-4 | 1e-4 | 1e-4 |
| # Iterations | 100 | 150 | 100 | 100 | 100 | 100 |
| # Epochs/Iteration | 20 | 20 | 20 | 20 | 20 | 20 |
| State Adversary $\|\xi_x\|_{\max}$ | 0.0 | 0.025 | 0.0 | 0.025 | 0.0 | 0.025 |
| Action Adversary $\|\xi_u\|_{\max}$ | 0.0 | 0.1 | 0.0 | 0.1 | 0.0 | 0.1 |
| Model Adversary $\|\xi_\theta\|_{\max}$ | 0.0 | 0.15 | 0.0 | 0.1 | 0.0 | 0.009 |
| Obs Adversary $\|\xi_o\|_{\max}$ | 0.0 | 0.025 | 0.0 | 0.025 | 0.0 | 0.025 |

## APPENDIX H    EVALUATIONS: RCI MODELING ERROR AND INITIAL CONDITION ABLATION STUDY

When benchmarking the characteristics of a new RL control framework, studies must address the central question: Does the RL algorithm deliver better performance than existing, well-established classical methods? In this evaluation, we provide substantive quantitative analysis demonstrating that RCI offers significant performance improvements over classical LQR. In short, in the face of severe modeling errors, for all systems RCI reliably delivers a 90% reduction in operator-norm error with respect to the optimal controller over a nominal LQR design. In the process of substantiating this claim, we also establish the key convergence and conditioning properties of RCI with respect to significant variations in 1) system initial conditions $x_0 \in G_{x_0}$ (50), and 2) modeling error $\nu \in G_\nu$ (53).

For a detailed discussion of the dynamical models, see Appendices D-F. All hyperparameter selections and definitions of the performance metrics examined in these studies can be found in Appendix G.

**Training Results.** Running RCI over the IC sweep $x_0 \in G_{x_0}$ for varying modeling errors $\nu \in G_\nu$, we plot the final-iteration controller error $e_{K_j}$ (54) over the sweep in Figure 5 (top row). In the bottom row of Figure 5, we examine the IC-sweep worst-case controller error $e_{K_j,\max}$ (57) as a function of modeling error $\nu$ (i.e., the max being taken over the initial condition grid $x_0 \in G_{x_0}$, cf. Appendix G for further details). Tables 18, 19, and 20 provide max, mean, and standard deviation data of the metrics presented in Figure 5 for the pendulum, jet, and DDMR, respectively.

**Remark H.1 (RCI Matches Optimal LQ Performance on Nominal Model, Delivers on Theoretical Guarantees)** Examining the nominal-model learning $\nu = 1$ in the top row of Figure 5 (blue curve) and Tables 18-20, we see that RCI successfully converges to the optimal controller for the nominal model $\nu = 1$ regardless of the IC chosen for all systems. Indeed, RCI exhibits a worst-case controller error $e_{K_1,\max} = 3.71 \times 10^{-5}$ for the pendulum system. For the jet aircraft, RCI has worst-case controller errors of $e_{K_1,\max} = 2.55 \times 10^{-15}$ in the translational loop $j = 1$ and $e_{K_2,\max} = 2.42 \times 10^{-8}$ in the rotational loop $j = 2$. For the DDMR, RCI has worst-case controller errors of $e_{K_1,\max} = 4.78 \times 10^{-9}$ in the translational loop $j = 1$ and $e_{K_2,\max} = 5.17 \times 10^{-5}$ in the

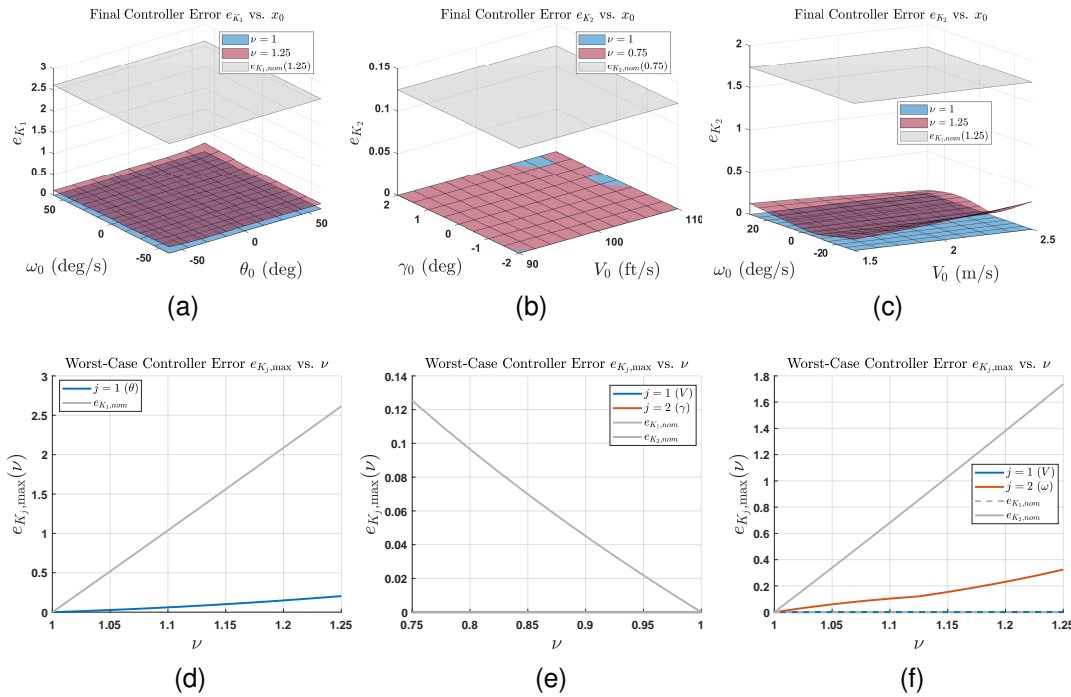

Figure 5: RCI modeling error $\nu$ and IC $x_0$ ablation study results. First column: pendulum. Second column: jet aircraft. Third column: DDMR. First row: Final controller error $e_{K_j}$ (54) versus IC $x_0 \in G_{x_0}$ (48). Second row: Worst-case controller error $e_{K_j,\max}(\nu)$ (57).

Table 18: RCI IC/modeling error sweep convergence performance – pendulum

| $\nu$ (24) | Loop $j$ | $e_{K_j,nom}$ (55) | Data Over $x_0 \in G_{x_0}$ | $e_{K_j}$ (54) | $\%e_{K_j}$ (56) |
|---|---|---|---|---|---|
| 1 | 1 | 0 | max | 3.71e-05 | N/A[a] |
| | | | avg | 1.75e-05 | N/A |
| | | | std | 1.19e-05 | N/A |
| 1.1 | 1 | 1.0383 | max | 0.0610 | 5.88 |
| | | | avg | 0.0337 | 3.25 |
| | | | std | 0.00547 | 0.527 |
| 1.25 | 1 | 2.6139 | max | 0.204 | 7.82 |
| | | | avg | 0.125 | 4.79 |
| | | | std | 0.0140 | 0.536 |

[a]Not applicable for the nominal model $\nu = 1$.

rotational loop $j = 2$. Thus, RCI matches the optimal performance of classical LQR when training on a nominal model and achieves real-world convergence performance in exact accordance with its theoretical guarantees (cf. Section 2).

**Remark H.2 (RCI Reduces Controller Error Relative to Nominal LQR Design by a Factor of Ten, Demonstrates Real-World Synthesis Guarantees)** Having proven that RCI successfully converges to the optimal controller when training on the nominal model $\nu = 1$, we now turn our attention to its convergence properties in the presence of modeling error $\nu \neq 1$. We examine the first row of Figure 5, comparing the RCI controller error $e_{K_j}$ (54) for a 25% modeling error (red curve) with the nominal LQ controller error $e_{K_j,nom}(1.25)$ ($e_{K_j,nom}(0.75)$ in the case of the jet) (55) at this same modeling error (gray curve). As can be seen qualitatively in Figure 5, for all three systems

Table 19: RCI IC/modeling error sweep convergence performance – jet aircraft

| $\nu$ (29) | Loop $j$ | $e_{K_j,nom}$ (55) | Data Over $x_0 \in G_{x_0}$ | $e_{K_j}$ (54) | $\%e_{K_j}$ (56) |
|---|---|---|---|---|---|
| 1 | 1 | 0 | max | 2.55e-15 | N/A[a] |
| | | | avg | 1.61e-15 | N/A |
| | | | std | 4.09e-16 | N/A |
| | 2 | 0 | max | 2.42e-08 | N/A[a] |
| | | | avg | 1.93e-08 | N/A |
| | | | std | 2.54e-09 | N/A |
| 0.9 | 1 | 0 | max | 2.66e-15 | N/A[b] |
| | | | avg | 1.49e-15 | N/A |
| | | | std | 3.76e-16 | N/A |
| | 2 | 0.04507 | max | 2.15e-08 | 4.77e-05 |
| | | | avg | 1.71e-08 | 3.79e-05 |
| | | | std | 2.29e-09 | 7.06e-06 |
| 0.75 | 1 | 0 | max | 2.27e-15 | N/A[b] |
| | | | avg | 1.26e-15 | N/A |
| | | | std | 3.29e-16 | N/A |
| | 2 | 0.12533 | max | 1.79e-08 | 1.43e-05 |
| | | | avg | 1.38e-08 | 1.10e-05 |
| | | | std | 1.94e-09 | 1.55e-06 |

[a]Not applicable for the nominal model $\nu = 1$.

[b]Modeling error $\nu$ does not affect optimal controller $K_1^*$ in loop $j = 1$ for the jet aircraft (cf. Appendix E.2).

RCI consistently achieves a significantly reduced controller error with respect to the nominal LQ design regardless of the IC chosen.

For the pendulum, Table 18 shows for a 25% modeling error $\nu = 1.25$ that RCI exhibits worst-case controller error $e_{K_1,\max}(1.25) = 0.204$. By contrast, the nominal controller error is $e_{K_1,nom}(1.25) = 2.6139$; thus, as a worst-case RCI delivers an optimality error $\%e_{K_1}$ (56) of only 7.82% relative to the nominal LQ design; i.e., a 92% reduction. For the jet, Table 19 shows for a 25% modeling error $\nu = 1.25$ that RCI exhibits worst-case controller error is $e_{K_2,\max}(1.25) = 1.79 \times 10^{-8}$ in the rotational loop $j = 2$, a 99% reduction from nominal. For the DDMR, Table 20 shows for a 25% modeling error $\nu = 1.25$ that RCI exhibits worst-case controller error is $e_{K_2,\max}(1.25) = 0.324$ in the rotational loop $j = 2$. By contrast, the nominal controller error $e_{K_2,nom}(1.25) = 1.7375$; thus, as a worst-case RCI delivers an optimality error $\%e_{K_2}$ (56) of only 18.65% relative to the nominal LQ design; i.e., an 81% reduction. On average, RCI offers a 94% reduction in optimality error in loop $j = 2$, with a standard deviation 3.23%.

Table 20: RCI IC/modeling error sweep convergence performance – DDMR

| $\nu$ (43) | Loop $j$ | $e_{K_j,nom}$ (55) | Data Over $x_0 \in G_{x_0}$ | $e_{K_j}$ (54) | $\%e_{K_j}$ (56) |
|---|---|---|---|---|---|
| 1 | 1 | 0 | max | 4.78e-09 | N/A[a] |
| | | | avg | 4.67e-09 | N/A |
| | | | std | 5.56e-11 | N/A |
| | 2 | 0 | max | 5.17e-05 | N/A[a] |
| | | | avg | 1.19e-05 | N/A |
| | | | std | 9.06e-06 | N/A |
| 1.1 | 1 | 0 | max | 1.03e-04 | N/A[b] |
| | | | avg | 1.03e-04 | N/A |
| | | | std | 3.31e-07 | N/A |
| | 2 | 0.68206 | max | 0.103 | 15.15 |
| | | | avg | 0.0610 | 8.94 |
| | | | std | 0.0277 | 4.07 |
| 1.25 | 1 | 0 | max | 3.08e-04 | N/A[b] |
| | | | avg | 3.05e-04 | N/A |
| | | | std | 9.56e-07 | N/A |
| | 2 | 1.7375 | max | 0.324 | 18.65 |
| | | | avg | 0.0982 | 5.65 |
| | | | std | 0.0563 | 3.23 |

[a]Not applicable for the nominal model $\nu = 1$.

[b]Modeling error $\nu$ does not affect optimal controller $K_1^*$ in loop $j = 1$ for the DDMR (cf. Appendix F.2).

## APPENDIX I   EVALUATIONS: QUANTITATIVE COMPARISONS BETWEEN RCI & FVIS

In this section, we examine the effects of modeling error $\nu$ on 1) policy performance $J$ (2), 2) critic network approximation error $J - V$, and 3) closed-loop performance. All hyperparameter selections can be found in Appendix G.

**RCI Weight Responses, cFVI/rFVI Learning Curves**. We first present convergence performance of RCI weights and the learning curves for the two deep RL methods in Figure 6. We generated these curves using an identical procedure to the original FVI studies (Lutter et al., 2023a; 2022) using the original authors's code (Lutter et al., 2023b), the results obtained over 20 seeds. As a note, due to RCI's classical structure, its learning is completely deterministic given a set of state-action data. Thus, RCI does not require random number generation, so training over seeds does not apply. As can be seen, RCI exhibits monotonic weight responses without chatter for all three systems.

### I.1   TRAINING RESULTS: COST PERFORMANCE

Figure 7 shows the cost difference $J_{cFVI} - J_{RCI}$ between cFVI and RCI (first row), and the difference $J_{rFVI} - J_{RCI}$ between rFVI and RCI (second row) for the nominal pendulum model $\nu = 1$ (left column), a 10% modeling error $\nu = 1.1$ (middle column), and a 25% modeling error $\nu = 1.25$ (right column). Note that wherever this difference is positive, RCI delivers *better* performance than the respective FVI algorithm. Table 21 presents the corresponding min, max, average, and standard deviation data. Figure 8 and Table 22 are laid out analogously for the jet aircraft, and Figure 9 and Table 23 are laid out analogously for the DDMR.

**Cost Performance – Pendulum.** In short, RCI and cFVI perform comparably overall, cFVI edging out RCI near the nominal model but degrading with increasing modeling error. By contrast, rFVI's performance is significantly worse than RCI's or cFVI's. Examining the first row of Figure 7, the RCI and cFVI policies deliver highly comparable cost performance $J$ in a large region around the origin for the nominal model $\nu = 1$ and 10% modeling error $\nu = 1.1$. Toward the fringes of the test domain, the performance of cFVI edges out that of RCI slightly, by as much as -0.243 (Table 21).

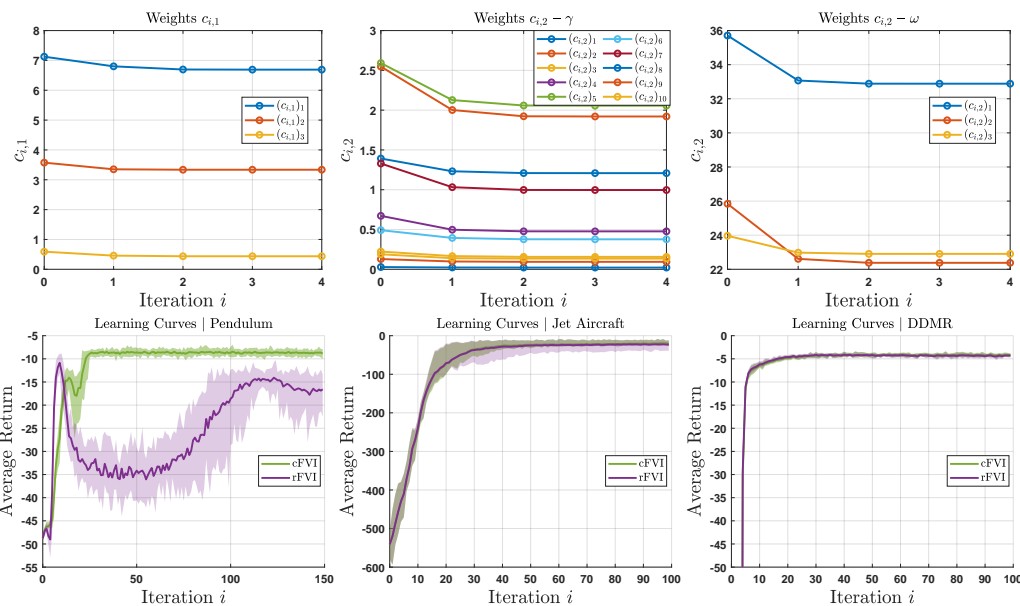

Figure 6: Convergence plots for pendulum (left), jet aircraft (middle), and DDMR (right). Top row: RCI weight responses $c_{i,j}$ (7) for IC $x_0 = x_e$ on nominal model $\nu = 1$. Bottom row: Learning curves of cFVI and rFVI, obtained over 20 seeds. The shaded area displays the min/max range between seeds, as is presented in the original works (Lutter et al., 2023a; 2022).

Specifically, cFVI performs better near the corners $x = (-60, -60)$ and $x = (60, 60)$; i.e., for large initial pendulum displacements and velocities in the direction of the displacement. This is perhaps intuitive, since these trajectories depart furthest from the origin. Here, the pendulum nonlinearities are strongest, and cFVI's deep network will have an approximation advantage over RCI's quadratic cost approximator. However, we note that is precisely in these two regions that the performance of cFVI degrades the heaviest in relation to RCI when the modeling error is increased. For a 25% modeling error $\nu = 1.25$, Figure 7c shows that in these corners the RCI policy is far superior, by as much as 0.499 at max (Table 21).

Meanwhile, the second row of Figure 7 shows that rFVI performs comparably to RCI and cFVI in near the origin, but its policy performance degrades significantly on the same fringes. Indeed, Table 21 shows that RCI exhibits lower cost pointwise relative to rFVI, and rFVI's worst-case cost increases from 6.50 at nominal to 8.59 at the 25% modeling error relative to RCI. Overall, the cost performance of RCI and cFVI are much more comparable, and rFVI exhibits the worst performance across the board.

**Cost Performance – Jet Aircraft.** For the jet aircraft, RCI perhaps enjoys the largest performance advantage over the FVIs of any of the systems tested. Examination of the jet cost data in Table 22 shows that RCI delivers the lowest cost pointwise regardless of modeling error. Furthermore, the cost discrepancy between the FVIs and RCI is the largest of the three systems tested, averaging at least 3.08 for cFVI and 3.72 for rFVI. rFVI's degradation is less severe than cFVI's, but rFVI delivers inferior cost performance overall, as much as 10.34 higher than RCI's at max. Meanwhile, examining Figure 8 shows that cFVI and rFVI exhibit similar cost performance behavior for this system. Both compare well with RCI for lower initial airspeeds $V$, but a large performance discrepancy develops at higher airspeeds. The discrepancy is seen to be slightly worse in the rFVI case.

**Cost Performance – DDMR.** Figure 9 shows the cost difference data for the DDMR. Note that RCI delivers lower cost $J$ than cFVI and rFVI *pointwise*, regardless of modeling error – an important result for training on this more complicated real-world DDMR system. The performance of cFVI in relation to RCI degrades with increasing modeling error, the maximum performance discrepancy $J_{cFVI} - J_{RCI}$ increasing by 50% from 0.273 on the nominal model $\nu = 1$ to 0.414 at $\nu = 1.25$ (cf. Table 23). On the other hand, rFVI's performance improves with modeling error, its max

performance discrepancy decreasing from 2.27 at $\nu = 1$ to 1.60 at $\nu = 1.25$. Nevertheless, as with the pendulum system, rFVI exhibits the worst overall cost performance of the three methods.

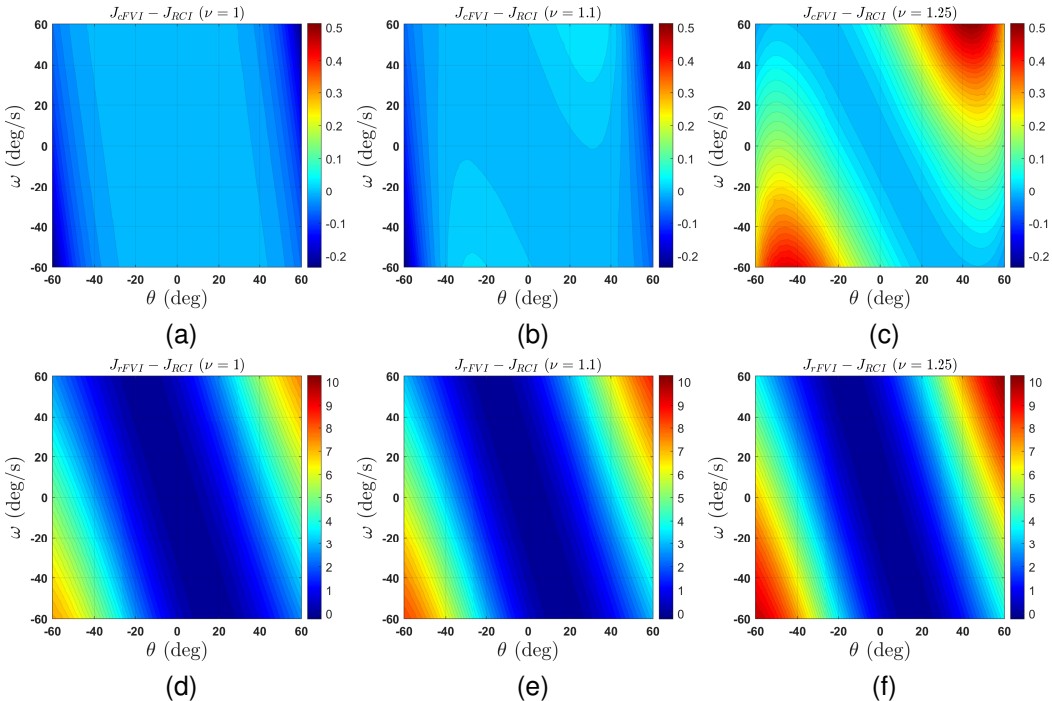

Figure 7: Cost performance results of pendulum model. First row: Cost difference $J_{cFVI} - J_{RCI}$ (2). Second row: Cost difference $J_{rFVI} - J_{RCI}$ (2). Left: Nominal model $\nu = 1$ (43). Middle: 10% modeling error $\nu = 1.1$. Right: 25% modeling error $\nu = 1.25$.

Table 21: Pendulum training cost/approximation data

| Function | Data | $\nu$ (24) | | |
|---|---|---|---|---|
| | | 1 | 1.1 | 1.25 |
| $J_{cFVI} - J_{RCI}$ | min | -0.230 | -0.235 | -0.0248 |
| | max | 4.628e-04 | 0.0384 | 0.512 |
| | avg | -0.0196 | -0.00950 | 0.122 |
| | std | 0.0352 | 0.0359 | 0.121 |
| $J_{rFVI} - J_{RCI}$ | min | -1.33e-06 | 2.43e-06 | -3.81e-4 |
| | max | 7.72 | 8.80 | 10.27 |
| | avg | 2.16 | 2.50 | 2.99 |
| | std | 1.92 | 2.20 | 2.61 |
| $J(x) - V(x)$ RCI | min | -1.758 | -2.28 | -3.24 |
| | max | 1.93e-04 | 0.00828 | 0.0431 |
| | avg | -0.257 | -0.325 | -0.444 |
| | std | 0.345 | 0.447 | 0.626 |
| $J(x) - V(x)$ cFVI | min | -0.0213 | -0.00732 | -0.00941 |
| | max | 0.0338 | 2.75 | 8.29 |
| | avg | -0.00407 | 0.627 | 1.88 |
| | std | 0.00839 | 0.610 | 1.83 |
| $J(x) - V(x)$ rFVI | min | -11.80 | -7.98 | -1.54 |
| | max | -0.04e-4 | -0.00142 | -0.00142 |
| | avg | -3.11 | -2.16 | -0.544 |
| | std | 2.81 | 1.93 | 0.436 |

Table 22: Jet aircraft training cost/approximation data

| Function | Data | $\nu$ (29) | | |
|---|---|---|---|---|
| | | 1 | 0.9 | 0.75 |
| $J_{cFVI} - J_{RCI}$ | min | 0.00 | 0.00 | 0.00 |
| | max | 8.04 | 7.90 | 8.57 |
| | avg | 3.08 | 3.25 | 3.74 |
| | std | 1.80 | 1.82 | 1.98 |
| $J_{rFVI} - J_{RCI}$ | min | 0.00 | 0.00 | 0.00 |
| | max | 10.15 | 10.23 | 10.34 |
| | avg | 3.72 | 3.86 | 4.22 |
| | std | 2.23 | 2.24 | 2.29 |
| $J(x) - V(x)$ RCI | min | 0.00 | 0.00 | 0.00 |
| | max | 11.77 | 13.01 | 15.66 |
| | avg | 3.15 | 3.43 | 4.01 |
| | std | 2.75 | 3.00 | 3.52 |
| $J(x) - V(x)$ cFVI | min | -0.461 | -0.141 | -5.90e-04 |
| | max | 31.29 | 32.33 | 35.82 |
| | avg | 8.28 | 8.72 | 9.81 |
| | std | 6.57 | 6.69 | 7.14 |
| $J(x) - V(x)$ rFVI | min | -3.44 | -3.12 | -2.42 |
| | max | 32.75 | 33.73 | 36.30 |
| | avg | 7.93 | 8.35 | 9.30 |
| | std | 7.16 | 7.26 | 7.56 |

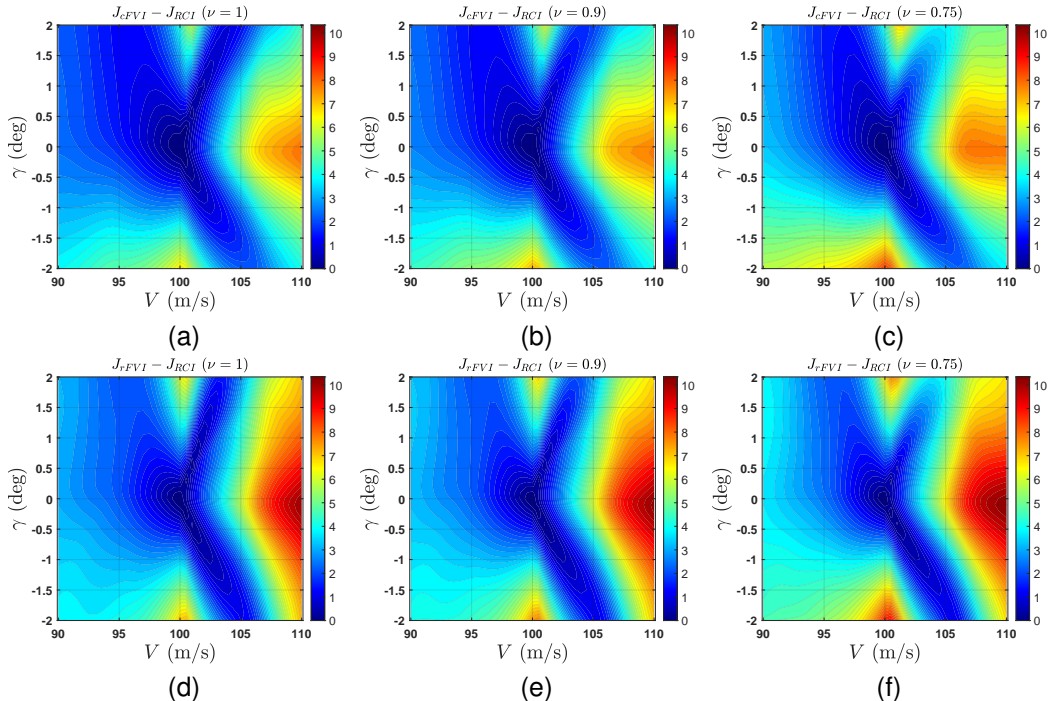

Figure 8: Cost performance results of jet aircraft model. First row: Cost difference $J_{cFVI} - J_{RCI}$ (2). Second row: Cost difference $J_{rFVI} - J_{RCI}$ (2). Left: Nominal model $\nu = 1$ (29). Middle: 10% modeling error $\nu = 0.9$. Right: 25% modeling error $\nu = 0.75$.

Table 23: DDMR training cost/approximation data

| Function | Data | $\nu$ (43) | | |
|---|---|---|---|---|
| | | 1 | 1.1 | 1.25 |
| $J_{cFVI} - J_{RCI}$ | min | 1.11e-05 | 1.07e-05 | 1.08e-05 |
| | max | 0.273 | 0.284 | 0.414 |
| | avg | 0.0912 | 0.0933 | 0.130 |
| | std | 0.0773 | 0.0792 | 0.112 |
| $J_{rFVI} - J_{RCI}$ | min | 0.00177 | 0.00177 | 0.00177 |
| | max | 2.27 | 1.99 | 1.60 |
| | avg | 0.630 | 0.562 | 0.462 |
| | std | 0.535 | 0.472 | 0.381 |
| $J(x) - V(x)$ RCI | min | -0.698 | -0.759 | -0.931 |
| | max | 0.824 | 0.990 | 1.147 |
| | avg | 0.00583 | 0.0197 | 0.0117 |
| | std | 0.201 | 0.231 | 0.274 |
| $J(x) - V(x)$ cFVI | min | -0.0128 | -0.00105 | -0.00105 |
| | max | 0.325 | 0.806 | 1.802 |
| | avg | 0.0961 | 0.241 | 0.498 |
| | std | 0.0904 | 0.157 | 0.376 |
| $J(x) - V(x)$ rFVI | min | -6.08 | -5.80 | -5.36 |
| | max | 3.15e-04 | 3.31e-04 | 3.57e-04 |
| | avg | -1.79 | -1.72 | -1.60 |
| | std | 1.30 | 1.23 | 1.13 |

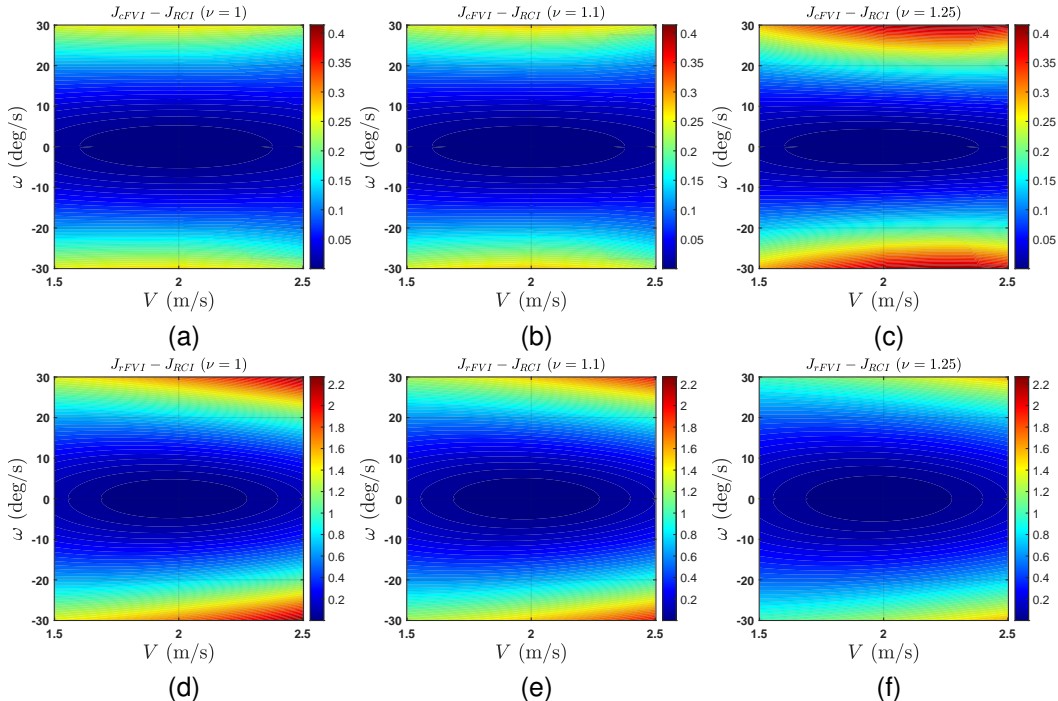

Figure 9: Cost performance results of DDMR model. First row: Cost difference $J_{cFVI} - J_{RCI}$ (2). Second row: Cost difference $J_{rFVI} - J_{RCI}$ (2). Left: Nominal model $\nu = 1$ (43). Middle: 10% modeling error $\nu = 1.1$. Right: 25% modeling error $\nu = 1.25$.

### I.2    TRAINING RESULTS: APPROXIMATION PERFORMANCE

Figures 10, 11, and 12 show the the critic network error $J - V$ for RCI (first row), cFVI (second row), and rFVI (third row) for the pendulum, jet, and DDMR systems, respectively. In general, it is desirable for the difference $J - V$ to be as small in magnitude as possible (so the critic is accurate) and to be negative if it does deviate from zero (so the critic underestimates the policy performance).

**Approximation Performance – Pendulum.** Examining Figure 12, the overall picture is clear: RCI exhibits the most consistent approximation performance, while the performance of the two FVI methods is much more sensitive to modeling error. RCI's network is highly accurate in a large region around the origin and slightly underestimates the policy performance toward the fringes. This underestimation increases in magnitude monotonically with the modeling error. However, the degradation is gradual, beginning at a worst-case approximation error of -1.758 for the nominal model $\nu = 1$ and decreasing to only -3.24 for a 25% modeling error $\nu = 1.25$ (Table 21).

cFVI's critic does an excellent job of approximating the policy cost for the nominal model $\nu = 1$, the two functions falling within 0.0338 of each other at max (Table 21). Given cFVI's deep network critic structure, such approximation performance is intuitive. However, with the introduction of a 10% modeling error $\nu = 1.1$, the critic approximation quality degrades significantly. Examining Table 21, cFVI overestimates its policy's performance by 0.627 on average, 2.75 at max for $\nu = 1.1$, compared to RCI's underestimation by only -0.325 on average, -2.28 at min for the same modeling error. Thus, for even mild modeling errors, RCI exhibits an underestimation behavior preferable to the overestimation behavior of cFVI, and in magnitude RCI's critic error is approximately half that of cFVI on average. We note that cFVI also tends to overestimate its policy performance for the DDMR system as well (see below), suggesting this is a common performance characteristic of the method. The discrepancies grow more pronounced as we move to the severe 25% modeling error $\nu = 1.25$. Here, RCI underestimates its policy performance by -3.24 at minimum, -0.444 on average. In comparison, cFVI overestimates its policy performance by 8.29 at max, 1.88 on average.

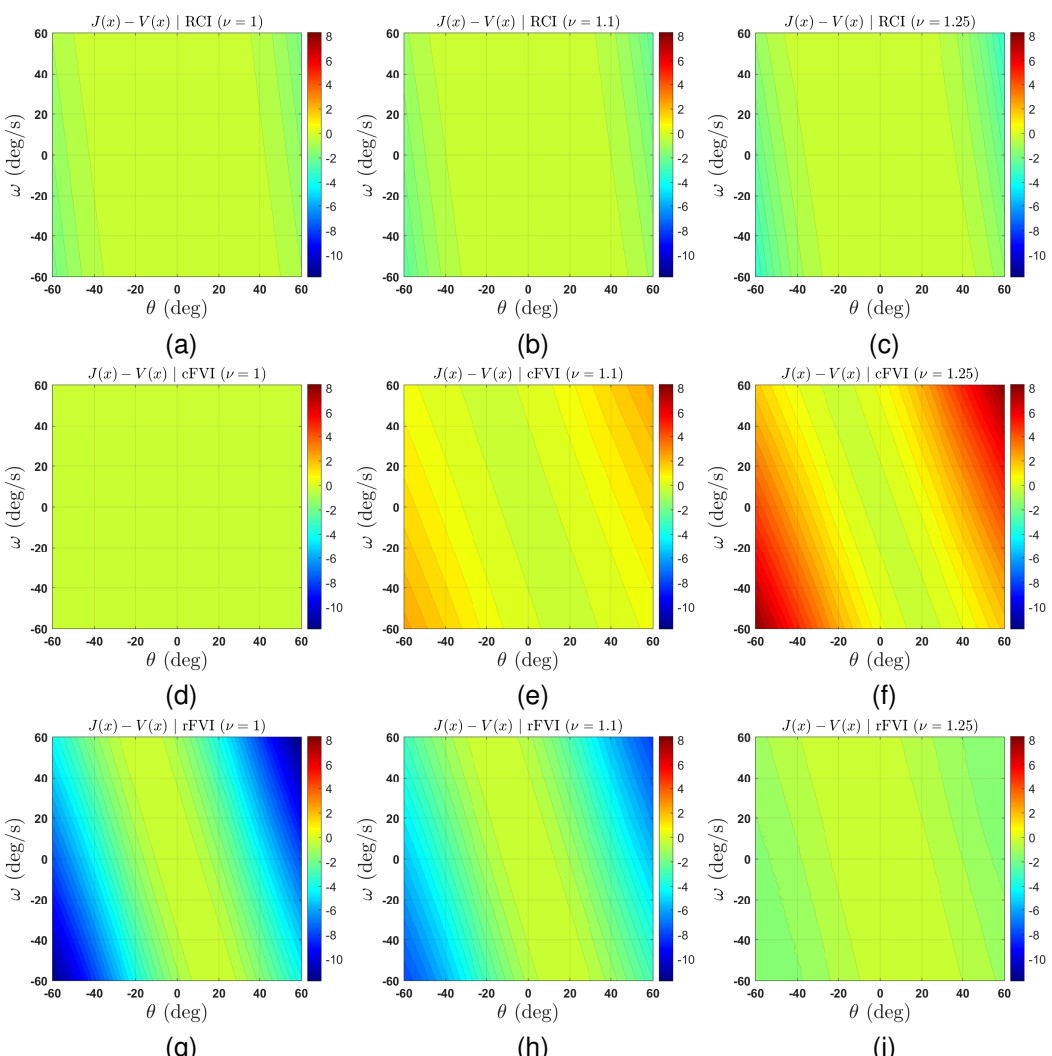

Figure 10: Critic NN approximation error $J(x) - V(x)$ of pendulum model. Left: Nominal model $\nu = 1$ (24). Middle: 10% modeling error $\nu = 1.1$. Right: 25% modeling error $\nu = 1.25$. First row: RCI. Second row: cFVI (Lutter et al., 2023a). Third row: rFVI (Lutter et al., 2022).

On the other hand, rFVI's approximation performance is poor for the nominal model and actually improves with increasing modeling error. rFVI underestimates its policy performance by -10.37 at worst for the nominal model, improving to -6.82 and -1.34 for 10% and 25% modeling errors, respectively. Thus, we conclude that rFVI's value function has successfully adapted to its modeling error adversary $\xi_\theta$, at the cost of inferior performance for models closer to the nominal.

**Approximation Performance – Jet Aircraft.** As with cost performance, for the jet aircraft RCI definitively surpasses the FVIs in approximation performance. Examination of Table 22 shows that the worst-case critic network error for RCI on the nominal model is only 11.77, compared to cFVI's 31.29 and rFVI's 32.75. Similar results hold when modeling error is introduced. The approximation performance seen visually in Figure 11 shows that RCI achieves low approximation error across the state domain. Meanwhile, similarly to the cost performance, cFVI and rFVI exhibit similar behavior and both struggle to approximate at higher airspeeds $V$.

**Approximation Performance – DDMR.** We display the critic approximation error $J - V$ of RCI, cFVI, and rFVI in Figure 12, laid out analogously to Figure 10. Overall, the trends are similar to those of the pendulum: RCI exhbits the most consistent approximation performance, while the per-

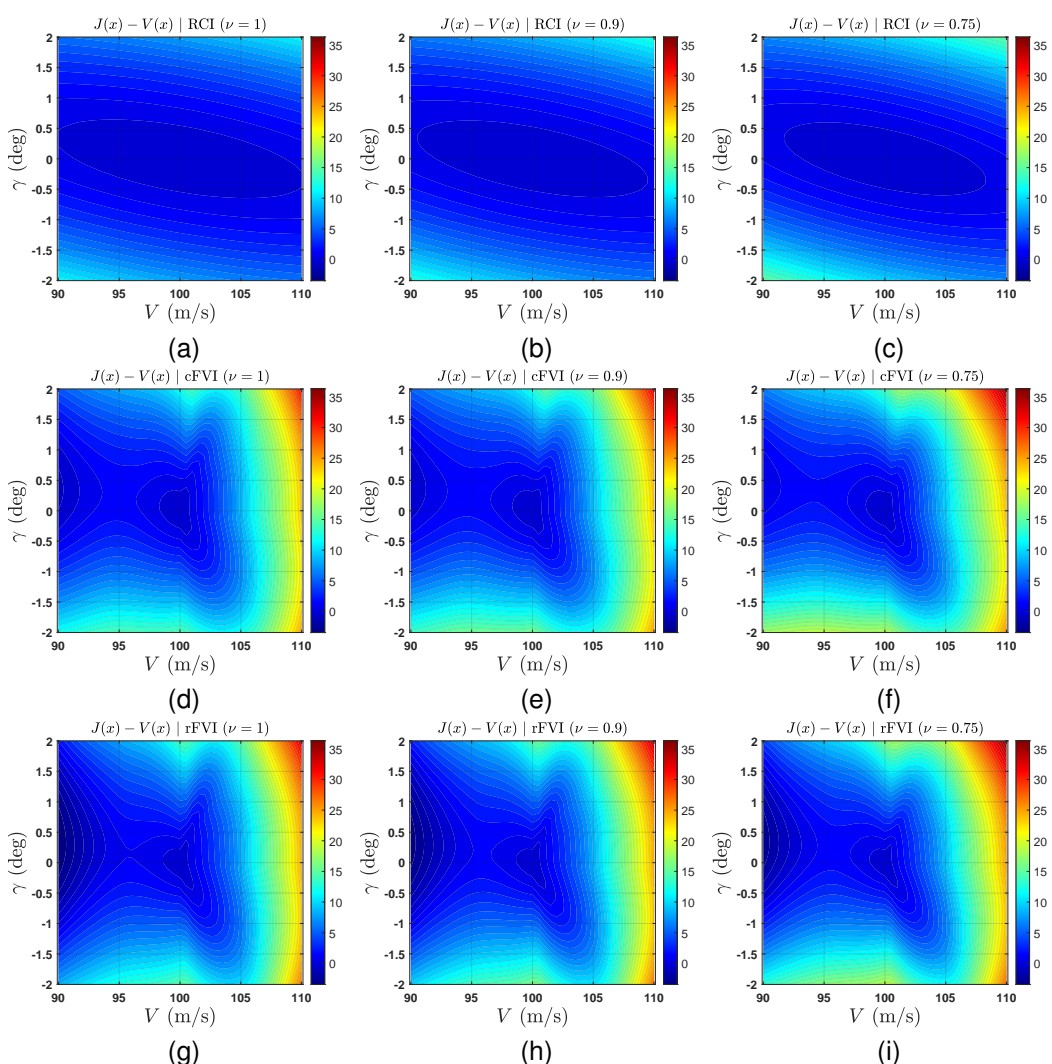

Figure 11: Critic NN approximation error $J(x) - V(x)$ of jet aircraft model. Left: Nominal model $\nu = 1$ (29). Middle: 10% modeling error $\nu = 0.9$. Right: 25% modeling error $\nu = 0.75$. First row: RCI. Second row: cFVI (Lutter et al., 2023a). Third row: rFVI (Lutter et al., 2022).

formance of the FVI methods is more sensitive to modeling error. cFVI does an excellent job of cost approximation in the case of the nominal model $\nu = 1$, edging out the approximation performance of RCI on the corners of the test grid. However, RCI exhibits much more consistent cost approximation performance in the face of modeling error. Indeed, as with the pendulum system, the approximation advantage of cFVI is lost when modeling error is introduced, eventually overestimating by up to 1.802 at max for the 25% modeling error to RCI's 1.147. Meanwhile, rFVI struggles the most with cost approximation, universally underestimating its policy performance by on the order of -6 at minimum.

## I.3 TRAINING RESULTS: CLOSED-LOOP PERFORMANCE

**Closed-Loop Performance – Pendulum.** We now study the swing-up performance of RCI, cFVI, the optimal LQ controller, and the nominal LQ controller (i.e., optimal for the nominal model $\nu = 1$). Figure 13a shows these responses for the nominal model $\nu = 1$, Figure 13b for a 10% modeling error $\nu = 1.1$, and Figure 13c for a 25% modeling error $\nu = 1.25$. As can be seen, the cFVI exhibits the most sluggish response, followed by rFVI. RCI and the LQ controllers are the most responsive.

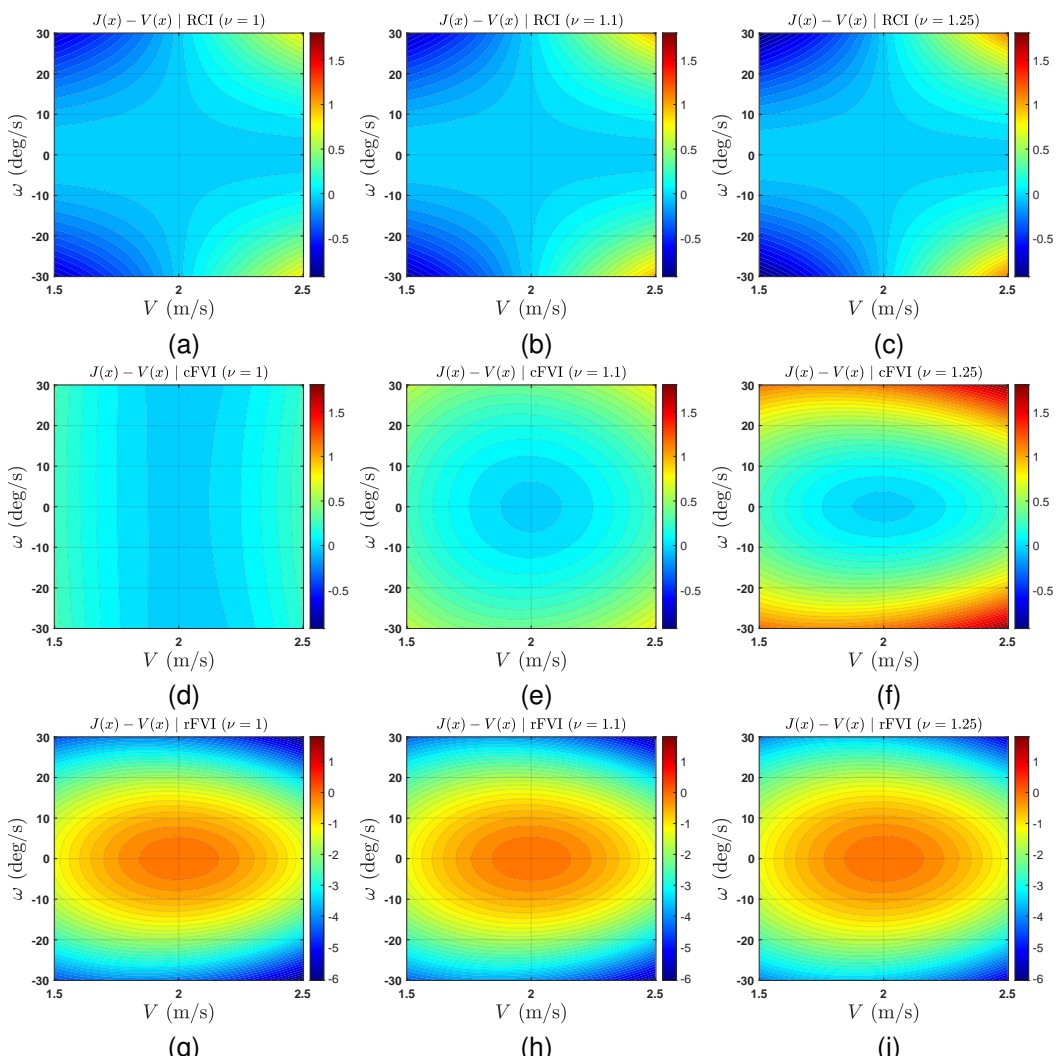

Figure 12: Critic NN approximation error $J(x) - V(x)$ of DDMR model. Left: Nominal model $\nu = 1$ (43). Middle: 10% modeling error $\nu = 1.1$. Right: 25% modeling error $\nu = 1.25$. First row: RCI. Second row: cFVI (Lutter et al., 2023a). Third row: rFVI (Lutter et al., 2022). Note: rFVI color normalized independently for legibility purposes.

cFVI/rFVI exhibit relatively slow closed-loop responses for the DDMR system as well (see below), suggesting that FVI tends to train to the control penalty more heavily. Regardless of the modeling error, RCI successfully recovers the closed-loop performance if the optimal LQ controller. As seen by the increased overshoot of the nominal LQ controller in Figures 13b and 13c, RCI successfully outperforms a classical LQR design in the face of modeling error.

**Closed-Loop Performance – Jet Aircraft.** Figure 14 plots the closed-loop responses to a 1 deg step FPA command. As can be seen, RCI exhibits a nice, monotonic response with no overshoot. By contrast, the FVIs exhibit a large overshoot transient and comparatively high settling time. The overshoot increases for both FVI algorithms as the modeling error increases.

**Closed-Loop Performance – DDMR.** Moving on to the DDMR, we study the closed-loop responses of RCI, cFVI, rFVI, the optimal LQ controller, and the nominal LQ controller to a 30 deg/s step angular velocity command in Figure 15. Figure 15a shows the responses for the nominal model $\nu = 1$, Figure 15b for 10% modeling error $\nu = 1.1$, and Figure 15c for 25% modeling error $\nu = 1.25$. As can be seen, regardless of the modeling error $\nu$, the closed-loop response of the cFVI

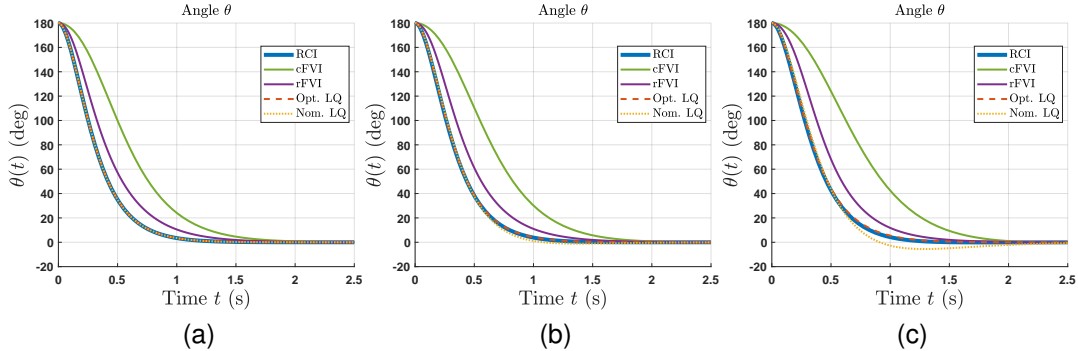

Figure 13: Swing-up closed-loop response of pendulum model. Left: Nominal model $\nu = 1$ (24). Middle: 10% modeling error $\nu = 1.1$. Right: 25% modeling error $\nu = 1.25$.

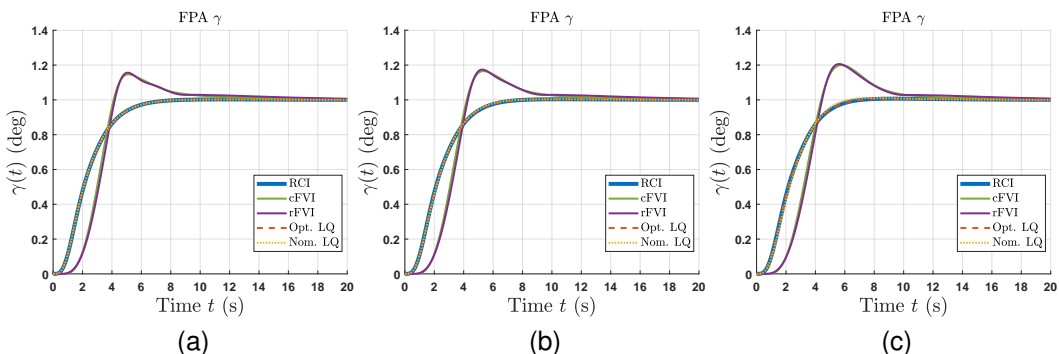

Figure 14: Closed-loop response of jet aircraft model to 1 deg step FPA command. Left: Nominal model $\nu = 1$ (29). Middle: 10% modeling error $\nu = 0.9$. Right: 25% modeling error $\nu = 0.75$.

controller has similar rise time to RCI, but with significant overshoot and slow settling time. The cFVI overshoot increases with increasing modeling error. Meanwhile, the rFVI response has similar overshoot to RCI, but with relatively long rise time and very sluggish settling time. Regardless of the modeling error, RCI successfully recovers the closed-loop performance of the optimal LQ controller. As seen by the increased overshoot of the nominal LQ controller in Figures 15b and 15c, RCI successfully outperforms a classical LQR design.

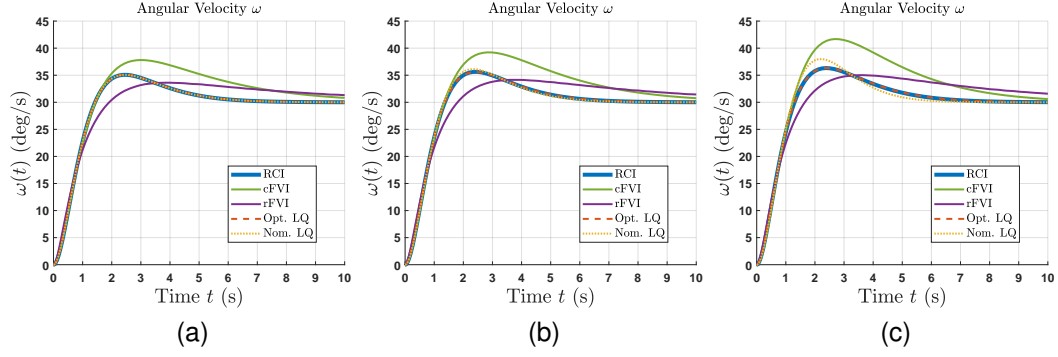

Figure 15: Closed-loop response of DDMR model to 30 deg/s step angular velocity command. (a): Nominal model $\nu = 1$ (43). (b): 10% modeling error $\nu = 1.1$. (c): 25% modeling error $\nu = 1.25$.

## APPENDIX J    ADP PERFORMANCE AND DESIGN INSIGHTS

ADP approaches were developed largely within the scope of seminal works such as integral reinforcement learning (IRL) (Vrabie & Lewis, 2009), synchronous policy iteration (SPI) (Vamvoudakis & Lewis, 2010), robust ADP (RADP) (Jiang & Jiang, 2014), and continuous-time value iteration (CT-VI) (Bian & Jiang, 2022). As a result of ADP's theoretical frameworks in adaptive and optimal control, Lyapunov arguments are available to prove qualitative properties including weight convergence and closed-loop stability results. However, the results require restrictive theoretical assumptions (see detailed list in Remark C.3) which are difficult to satisfy for even simple academic examples. Furthermore, the methods do not provide constructive design procedures for ensuring that the required hypotheses are met. In all, these restrictions result in significant numerical issues which limit ADP synthesis capability (Wallace & Si, 2022).

Besides the common limitations discussed above, each of the central four algorithms has individual numerical challenges exhibited empirically in Wallace & Si (2022). IRL's excitation quality quickly degrades as the state is regulated to the origin, since IRL does not accommodate probing noise excitation in its formulation (Vrabie & Lewis, 2009). SPI's gradient descent tuning laws often experience weight freezing due to a lack of PE and poor scaling of the required update terms. Meanwhile, RADP is sensitive to over-excitation and ultimately fails to stabilize the inverted pendulum. Finally, CT-VI's dynamic tuning laws require nested integration and matrix inversion which prove particularly susceptible to weight divergence.

The elegant results of ADPs have made significant contributions to CT-RL. However, further algorithm development work is required to enable these algorithms to synthesize for meaningful applications.

