# OpenReview forum: "A New, Physics-Based Continuous-Time Reinforcement Learning Algorithm with Performance Guarantees"
_ICLR.cc/2024/Conference — Submitted to ICLR 2024_

### Official Review · Reviewer_SQkk · 2023-10-24

**Soundness:** 2 fair
**Presentation:** 2 fair
**Contribution:** 2 fair
**Rating:** 3
**Confidence:** 4

**Summary:**

This paper presents a new continuous-time reinforcement learning (CTRL) algorithm for control of affine nonlinear systems. The key idea is to use reference command input (RCI) as probing noise in learning. The simulations show RCI leads to better results than fitted value iteration.

**Strengths:**

This paper has a good review of the existing ADP methods.

**Weaknesses:**

1. The methodology introduced in this paper is an extension of the RADP method, with the primary modification being the linearization of the nonlinear system. However, the implications of such linearization are not distinctly outlined, nor is there a clear comparative analysis with the traditional RADP method. The absence of a detailed examination of the linearization's impact raises questions about the method's efficacy and novelty.

2. The authors suggest that the rationale behind employing the RCI framework is its potential to enhance the PE condition. Nevertheless, the explanation as to why this approach is effective is insufficiently substantiated. Furthermore, the connection between the RCI and the employed linearization technique is ambiguous, resulting in a fragmented logical flow in the methodology's presentation.

3. The proposed methodology presupposes a comprehensive understanding of system dynamics. However, with known system dynamics, one could conduct policy iteration directly using a "differential" formulation as opposed to the "integral" formulation, which seems unnecessarily convoluted. For instance, a comparison could be made with the "Relaxed Actor-Critic" method detailed in [1], which offers a solution to the HJB equation through policy iteration in the context of fully understood system dynamics.

Reference: [1] J. Duan et al., "Relaxed Actor-Critic With Convergence Guarantees for Continuous-Time Optimal Control of Nonlinear Systems," in IEEE Transactions on Intelligent Vehicles, vol. 8, no. 5, pp. 3299-3311, May 2023, doi: 10.1109/TIV.2023.3255264.

4. Unfortunately, the link provided for the open-source code corresponding to the paper's methodology is inaccessible, which hinders peer verification and replicability of the results presented.

5. The proof presented for Theorem 2.1 is unconvincing. It employs the Closed-Loop Stability attribute of Kleinman’s Algorithm, but the narrative fails to clarify why this particular inference is applicable to nonlinear systems as well. The proof lacks a thorough explanation, making the applicability of Kleinman’s Algorithm to nonlinear systems questionable.

**Questions:**

Why use the proposed method using linearization? What is the intuition behind it?

---

> ### Author Response · Authors · 2023-11-20
> **Response to Reviewer**
>
> This Reviewer's claims are addressed in the General Points of our rebuttal, and in our responses to the other two Reviewers. Please review.

---

> ### Comment · Reviewer_SQkk · 2023-11-22
> **Thank you for you response**
>
> Thank you very much for the explanation! It helps me a lot to understand the paper in depth.
>
> I can see my comments are similar to those of Reviewer QT7M.
>
> However, I still find it confusing and struggle to accept the claimed contributions, because compared to methods like RADP and Integral RL, the approach in this paper requires knowledge of system dynamics. From my understanding, when system dynamics are known, we can directly solve ADP using differential forms, utilizing the system's derivative information. Additional derivative information can make differential-form ADP more stable and can be directly solved using neural networks. The authors did not discuss this aspect, nor did they provide a performance comparison. I still believe that the ADP method with known system dynamics needs at least a comparison with other methods that also use known system dynamics, otherwise the performance comparison is not fair. I can accept that your method might be inferior to differential-form results, but I would like to see some discussion and experimental comparisons.
>
> Additionally, from a theoretical perspective, I still find the proof of algorithmic convergence provided by the authors to be tricky, at least not convincing enough for me.
>
> I see the effort put into this work and the authors' code is very clean, which is commendable. However, I still hope the authors will carefully consider the issues I've mentioned. Therefore, I do not recommend acceptance for publication this time.

---

> > ### Author Response · Authors · 2023-11-22
> > **(1 of 2) Thank you for the follow up comments. This has been an interesting exchange. Thank you for that. But, we are perplexed by where the Reviewer’s strong beliefs come from, and are backed up by what evidence...**
> >
> > > Thank you very much for the explanation! It helps me a lot to understand the paper in depth. I can see my comments are similar to those of Reviewer QT7M.
> >
> > * You are welcome, we are glad to provide some necessary backdrops on basic RL/ADP/control concepts.
> >
> > > However, I still find it confusing and struggle to accept the claimed contributions, because compared to methods like RADP and Integral RL, the approach in this paper requires knowledge of system dynamics.
> >
> > * Firstly: We encourage the Reviewer to GLANCE at how the IRL and RADP methods you argue for fare empirically in prior studies (Wallace \& Si, 2022). These both fail to synthesize for the toy problem:
> >
> >     \begin{align}
> >         f(x)
> >         =
> >         \left[
> >         \begin{array}{c}
> >              - x_{1} + x_{2} + 2 x_{2}^{3}
> >             \\\\
> >              - \frac{1}{2} (x_{1} + x_{2}) + \frac{1}{2} x_{2} (1 + 2 x_{2}^{2}) \sin^{2}(x_{1})
> >         \end{array}
> >         \right],
> >         \qquad
> >         g(x)
> >         =
> >         \left[
> >         \begin{array}{c}
> >              0
> >              \\\\
> >              \sin(x_{1})
> >         \end{array}
> >         \right].
> >     \end{align}
> >
> > * Secondly: As we have now stated in:
> >
> >    * Table 1 of the manuscript
> >    * Remark 2.1 of the manuscript
> >    * Our General Points 2 of this rebuttal
> >
> >   our proposed RCI work requires SIGNIFICANTLY LESS system dynamics knowledge than the leading FVI works in deep RL.
> >
> > * Allow us to copy and paste here again the "system dynamics requirement":
> >
> >   * **RCI:** $f, g$
> >   * **SOTA Deep RL FVIs:** $f, g$, $\partial f/\partial x$, $\partial g/\partial x$, $\partial f/\partial \theta$, $\partial g/\partial \theta$
> >
> > * Should we tell the greatly-successful deep RL FVI works that their results aren’t SOTA, then?
> >
> > > From my understanding, when system dynamics are known, we can directly solve ADP using differential forms, utilizing the system's derivative information. Additional derivative information can make differential-form ADP more stable and can be directly solved using neural networks.
> >
> > * Yes, and No -- 1) we are glad that the reviewer is aware that the Hamilton-Jacobi-Bellman equation (HJB) PDE offers an exact solution to optimal control problems when the dynamics are known. 2) However, except for toy problems, "directly" solving the HJB as you propose is numerically intractable. Hence the inception of **approximate** dynamic programming (ADP). By the way, there is a quick timeline in the development of ADP/RL for optimal control in (Wallace \& Si, 2023).
> >
> > * We would like to point out to this reviewer that the above toy problem was constructed such that the closed-form HJB solution is known *a priori* (because it is an unrealistic toy problem) in exact polynomial closed-form:
> >
> >     \begin{align}
> >         V^{*}(x) = \frac{1}{2} x_{1}^{2} + x_{2}^{2} + x_{2}^{4}
> >     \end{align}
> >
> >    Yet still, IRL and RADP fail to synthesize this $V^{*}$.
> >
> > * **This is a venue for SOTA learning results, not already-published findings.** The "experimental comparisons" you seek for ADP are already well documented in the literature.
> >
> > * **More on the “differential form” approach:** Please note that, contrary to the Reviewer’s opinion, we are not sure if the work of Duan et al. can be considered SOTA yet for the following reasons:
> >
> >    1. Evaluations: Its evaluations are far less than what is typically required and presented in leading venues such as ICLR.
> >    2. Solution Method: As with all ADP/RL algorithms, these methods still solve the HJB approximately. There need to be more comprehensive and substantial evaluations to solidify its performance characteristics.
> >
> > * **More on why we consider FVI SOTA:** Also contrary to the reviewer’s belief, we consider FVI results SOTA for the following reasons:
> >
> >    1. Evaluations: This work presents substantial evaluations on three SOTA CT-RL environments, including generalization studies with respect to system initial conditions and modeling error.
> >    2. Solution Method: FVI is a time-tested method in the DT setting, and (see Lutter et al., 2023a, 2022) this CT formulation inherits many of the favorable properties in DT.
> >
> > * **In short about this SOTA method discussion:** We understand the Reviewer’s skepticism, and we also understand the Reviewer’s endorsement of "differential form" approach. We would like to bring to the Reviewer’s attention that our "integral" reinforcement method has been a standard ADP learning principle for almost 15 years (Vrabie \& Lewis, 2009). Instead, the "differential" method (J. Duan et al., 2023) has yet to be proven by substantial evaluations to the standard of highest quality venues such as the ICLR.

---

> > > ### Author Response · Authors · 2023-11-22
> > > **(2 of 2) Thank you for the follow up comments. This has been an interesting exchange. Thank you for that. But, we are perplexed by where the Reviewer’s strong beliefs come from, and are backed up by what evidence...**
> > >
> > > > The authors did not discuss this aspect, nor did they provide a performance comparison.
> > >
> > > * **We discussed all of these points directly on page 1 of the manuscript.** See the second paragraph, where we even boldface **ADP**.
> > >
> > > * We furthermore devote an entire appendix (Appendix J) to discussing these performance limitations of ADP in greater detail, and we refer the reader there on page 1 for more insights.
> > >
> > > * Please read the manuscript.
> > >
> > > > I can accept that your method might be inferior to differential-form results, but I would like to see some discussion and experimental comparisons.
> > >
> > > * As far as your assertion that our results are "inferior" to these ADP works, we encourage you to review the studies and discussions already mentioned to assess how this argument holds up empirically.
> > >
> > > * We are curious where the reviewer’s opinion that "RCI is inferior to ADP" comes from and is based on what evidence?
> > >
> > > * **Concluding on the differential methods:** With all that has been said, we do give the approach its deserving credit. But please review our discussion above for how we select SOTA results. We hope the authors of the differential methods publish more results in the future using SOTA benchmarks to justify its SOTA status.
> > >
> > > > I still believe that the ADP method with known system dynamics needs at least a comparison with other methods that also use known system dynamics, otherwise the performance comparison is not fair.
> > >
> > > * We agree, and we did. Please see above. Additionally, to make our comparisons fair and provide true SOTA results, we instead developed an in-depth quantitative evaluation with the leading **model-based** deep RL FVI methods (Lutter et al., 2023a, 2022), which include substantial evaluations with very detailed descriptions for duplicating their results. And we actually are able to duplicate their results.
> > >
> > > * Please review the basic features of these algorithms.
> > >
> > > > I still find the proof of algorithmic convergence provided by the authors to be tricky, at least not convincing enough for me.
> > >
> > > * Right now this is speculation. We are wondering if the reviewer has a mathematically-tenable argument to make? Could you provide a counterexample?

---

> > > > ### Comment · Reviewer_SQkk · 2023-11-23
> > > >
> > > > Thank you very much for the author's prompt reply. This is my last sincere attempt to impart knowledge of ADP to the author.
> > > >
> > > >
> > > > Firstly, in the context of continuous-time ADP, the solution of PEV in its differential form is just as mainstream as the integral form. If the authors are skeptical, they can refer to Chapter 9 in the textbook [1] and Section 3 in the review by Sutton [2]. For instance, in Section 3 of [2], the author divides PI into differential PI and integral PI.
> > > >
> > > > I am quite familiar with the review by Wallace & Si, 2022 that you mentioned, but its focus is on integral PI. To my understanding, the biggest advantage of Integral PI is that it does not require knowledge of internal dynamics, which is repeatedly mentioned in Lewis's original paper. However, your method requires knowledge of internal dynamics, making the comparison inherently unfair.
> > > >
> > > > Secondly, if you insist that only integral PI is mainstream, then I would also like you to recognize what the real SOTA algorithms in top-tier computer conferences are. Please refer to [3][4] from major ML venues on model-based CTRL that you have not mentioned.
> > > >
> > > >
> > > > [1] Liu, Derong, et al. Adaptive dynamic programming with applications in optimal control. Berlin: Springer International Publishing, 2017.
> > > >
> > > > [2] Lee, Jaeyoung, and Richard S. Sutton. "Policy iterations for reinforcement learning problems in continuous time and space—Fundamental theory and methods." Automatica 126 (2021): 109421.
> > > >
> > > > [3] Yildiz, Cagatay, Markus Heinonen, and Harri Lähdesmäki. "Continuous-time model-based reinforcement learning." International Conference on Machine Learning. PMLR, 2021
> > > >
> > > > [4] Holt, Samuel, et al. "Neural Laplace Control for Continuous-time Delayed Systems." International Conference on Artificial Intelligence and Statistics. PMLR, 2023.

---

### Official Review · Reviewer_UKDb · 2023-10-28

**Soundness:** 2 fair
**Presentation:** 1 poor
**Contribution:** 2 fair
**Rating:** 5
**Confidence:** 2

**Summary:**

This work introduces physics-based CT-RL algorithm for affine systems using reference command input.  It aims at providing theoretical guarantees while showing good performance.

**Strengths:**

1. Careful comparisons and evaluations (if the presentations become better, those should become clearer)

Theorem 2.1 could be a potential strength; but I could not quite follow the details here.
To be honest, it was very hard to parse the overall algorithm.
Why for nonlinear systems the policy K is introduced in the algorithm?  The author also mention mu as a policy.
Proposition A.1 is referred at several places but without clear connections.
For nonlinear systems, the results should only be satisfied locally?
I may be missing something here, but I believe improving presentations should largely help clarifying the strength of the theoretical statements.

**Weaknesses:**

1. From the cost 2, the system must stabilizes on a zero cost point and stays there without control input so that the cost exists: Although there is a comparison to other methods, I honestly think this is a strong assumption for practical purposes that this work claims to target.
2. The presentation is not well structured; perhaps it is better to present a conceptual procedures first with figures, pseudo algorithm etc., and then go into the details.  The authors also use some notations and concepts and describe them later; which make it harder to track; those should be mentioned at the conceptual presentation stage.
Also for experimental sections, I guess it is because of page limit, it is a bit hard to parse what is going on (no indent, no new line...).
3. More explanations around A, B (nominal linearization terms that are known) are needed.
4. Table 2 is hard to parse.  Table 4 could be improved to show which case works better for RCI.
5. For all of the tables (and some figures) in the appendix, they should have more descriptions in the captions and they could be improved so that it becomes easier to get the ideas.

**Questions:**

1. I don’t get what “Thus, RCI can improve learning of existing CT-RL algorithms” mean from the paragraph.  Can you elaborate on this?

---

> ### Author Response · Authors · 2023-11-20
> **Response to Reviewer**
>
> > From the cost 2, the system must stabilizes on a zero cost point and stays there without control input so that the cost exists: Although there is a comparison to other methods, I honestly think this is a strong assumption for practical purposes that this work claims to target.
>
> * We greatly appreciate the Reviewer's effort in reading our paper.
>
> * Respectfully, your assertion that ``the system must stabilizes on a zero cost point and stays there without control input so that the cost exists" is simply incorrect: The system need not reach the equilibrium in finite time for the cost to be finite. These sorts of considerations are addressed in any standard optimal control text; see, e.g., Frank Lewis's "Optimal Control."
>
> * Q-R cost (2) is far from a "strong assumption" -- In control problems it is **the standard** cost structure that has been assumed in optimal control all the way back to its inception with Kalman in the 1960s.
>
> * Q-R cost is also the predominant cost structure chosen by leading ADP CT-RL algorithms; see, e.g., (Vrabie \& Lewis, 2009), (Vamvoudakis \& Lewis, 2010), (Jiang \& Jiang, 2014), (Bian \& Jiang, 2022). The deep RL FVI works (Lutter et al., 2023a), (Lutter et al., 2022) also accommodate Q-R cost.
>
> * General RL settings oftentimes can accommodate different and flexible reward functions. We emphasize that we are addressing a continuous-time RL **dynamical control** problem, not a discrete decision making/reward problem.
>
> > For nonlinear systems, the results should only be satisfied locally? I may be missing something here [...]
>
> * As stated in Theorem 2.1: The convergence, optimality, and closed-loop stability results of RCI apply to the full nonlinear system (1).
>
> * These results are not global in nature, which is **by design.** We would kindly like to remind this Reviewer that global asymptotic stability results for nonlinear systems generally require extremely stringent theoretical assumptions on both the structure of the environment dynamics and the knowledge of the dynamics. For instance, we discuss the theoretical assumptions required for the global results of the ADP CT-VI method (Bian \& Jiang, 2022) in detail in Appendix Remark C.3 (pp. 16 of manuscript). These include:
>
>   * Existence and uniqueness of solutions to an uncountable family of finite-horizon HJB equations
>   * Properness of each solution to the finite-horizon HJB equation
>   * Convergence of family of solutions of finite-horizon HJB equation to the infinite-horizon HJB solution
>   * Invariance of closed-loop state/action trajectory to compact set with respect to the probing noise $d$
>   * Initial *globally asymptotically stabilizing* policy
>   * PE assumption on various learning signals
>   * Chosen basis functions approximate optimal value and its gradient uniformly on compact sets
>   * Chosen basis functions approximate optimal policy uniformly on compact sets
>   * Chosen basis functions approximate optimal Hamiltonian uniformly on compact sets
>   * Basis functions for critic network are linearly-independent
>   * Basis functions for actor network are linearly-independent
>   * Basis functions for Hamiltonian network are linearly-independent
>
> * We emphasize: The proposed method requires **not one** of these above theoretical assumptions.
>
> > I don’t get what "Thus, RCI can improve learning of existing CT-RL algorithms" mean from the paragraph. Can you elaborate on this?
>
> * In regards to this question, the associated Equation (13) (pp. 4) provides a direct means to accommodate reference command inputs for standard-formulation CT-RL algorithms. Thus, the empirical issues observed for CT-RL algorithms which only insert probing noise (Wallace \& Si, 2022) can be overcome by applying reference command input via (13).

---

> > ### Comment · Reviewer_UKDb · 2023-11-21
> > **Thank you for the response**
> >
> > Thank you for the responses;
> >
> > 1. I did not mean "reach ... in [finite time]" by "stabilizing".
> > 2. The cost is indeed standard if it is linear systems.  For nonlinear system that may not stabilize with certain control policy, there should be discount factor or other forms of cost.
> > 3. What confused me about local vs global is about the claims of Theorem 2.1.  It is partially due to the presentation, but it says about optimality of the solutions for nonlinear system although the original Kleinman talks about it for linear systems.  Can you elaborate more on this?
> >
> > Other parts are satisfactory to me; thank you

---

> > > ### Author Response · Authors · 2023-11-21
> > > **Thank you for your feedback. Please review standard results in optimal control/MDPs/RL.**
> > >
> > > > 1. I did not mean "reach ... in [finite time]" by "stabilizing".
> > >
> > > We thank the reviewer for the follow up questions.
> > >
> > > * Understood, and to be clear we didn't take your language "reach ... in [finite time]" to mean stabilizing the closed-loop system either. Our above response addresses your original assertion, which pertains to the existence of the cost (2): The system need not reach the equilibrium in finite time in order for the cost (2) to "exist" (i.e., be finite).
> > >
> > > > 2. The cost is indeed standard if it is linear systems. For nonlinear system that may not stabilize with certain control policy, there should be discount factor or other forms of cost.
> > >
> > > * Q-R is standard for both linear and nonlinear systems. Nonlinear stability results for non-discounted cost can be found in a number of standard texts and in a variety of learning settings, for example:
> > >
> > >    * In optimal control: Frank Lewis's "Optimal Control"
> > >    * In MDPs: Dimitri Bertsekas's "Neurodynamic Programming"
> > >    * In RL: Andrew Barto and Richard S. Sutton's "Reinforcement Learning: An Introduction"
> > >
> > > * Again, our Q-R cost structure **(including the lack of discounting)** is the **identical** cost structure used on nonlinear systems in SOTA ADP and deep CT-RL works.
> > >
> > > > 3. What confused me about local vs global is about the claims of Theorem 2.1. It is partially due to the presentation, but it says about optimality of the solutions for nonlinear system although the original Kleinman talks about it for linear systems. Can you elaborate more on this?
> > >
> > > * Please first note that in a RL setting, except for simple academic examples, the
> > > value and policy networks cannot achieve exact solutions of the optimal value and policy. Instead, approximate value and approximate policy are obtained through learning (and using some available knowledge of system dynamics).
> > >
> > > * Thus, for RL algorithms optimality concerns the best approximation achievable given the neural networks used in approximating the value function (i.e., as it is stated in Theorem 2.1). As shown in Theorem 2.1, RCI converges to Kleinman’s solution. That means that this is the optimal approximation achievable by the RCI algorithms using the learned weights $c_{i}$ and the choice of bases.
> > >
> > > > Other parts are satisfactory to me; thank you
> > >
> > > * Thanks again to this Reviewer!

---

> > > > ### Comment · Reviewer_UKDb · 2023-11-22
> > > > **Thank you for your response again**
> > > >
> > > > Thank you for your response again.
> > > > I wanted to let you know that I have read the response.
> > > > It might be partially due to my understanding, but I still believe the current presentation is a bit confusing.
> > > > I keep my score because of it, but I appreciate the authors' clarifications on the work.

---

> > > > > ### Author Response · Authors · 2023-11-22
> > > > > **This has been an interesting exchange. We thank the Reviewer for reading our paper.**
> > > > >
> > > > > > It might be partially due to my understanding, but I still believe the current presentation is a bit confusing. I keep my score because of it [...]
> > > > >
> > > > > * Thank you for your feedback through the rebuttal process, we have enjoyed the discussion.
> > > > >
> > > > > * While our conversation to this point has focused on the veracity of (basic) RL/control concepts, it would be even more interesting if we had a chance to discuss the specific features of our algorithm and its significance.
> > > > >
> > > > > * We are always receptive to improving our writing. It would have been even more helpful for the improvement of our writing had we received more specific questions on the algorithms and results directly.

---

### Official Review · Reviewer_QT7M · 2023-10-31

**Soundness:** 2 fair
**Presentation:** 2 fair
**Contribution:** 2 fair
**Rating:** 3
**Confidence:** 3

**Summary:**

The authors proposed a new method exclusively for solving LQR problems (restricted to Q-R cost functionals without cross terms) by leveraging input/output insights and the underlying control problem structure. This enables the proposed method to have theoretical foundation which is currently lacking in more general purpose methods including ADP and DeepRL. In several benchmark tasks, the proposed method outperforms or matches existing practice.

**Strengths:**

The authors specifically studied an important class of control problem, namely the affine nonlinear LQR problem, in continuous time. By leveraging the linear-quadratic property of the underlying problem structure, and utilizing Kleinman's method, the authors arrived at a theoretical guarantee unsurprisingly. The proposed method indeed outperform in tasks where underlying dynamics are known and deterministic.

**Weaknesses:**

The study of linear-quadratic problems has formed a long list, while the manuscript only mentioned a few general-purpose methods such as ADP and FVI. The weakness of this work hence can be summarized as follows.

1. This work failed to mention other similar works in continuous-time LQR setting where different exploitation of the same linear-quadratic structure (as Kleinman's method) leads to different theoretical guarantees and efficient algorithms. The authors may want to conduct a thorough survey on existing works and compare their approaches with other model-based continuous-time LQR methods. A few examples can be found like:

[1] Jeongho Kim, Jaeuk Shin, and Insoon Yang. Hamilton-jacobi deep q-learning for deterministic continuous-time systems with lipschitz continuous controls. The Journal of Machine Learning Research, 22(1):9363–9396, 2021.

[2] Haoran Wang, Thaleia Zariphopoulou, and Xun Yu Zhou. Reinforcement learning in continuous time and space: A stochastic control approach. The Journal of Machine Learning Research, 21(1): 8145–8178, 2020.

2. It is questionable if the method in this work can be fairly compared to other general purpose RL methods or ADP methods, since the latter typically won't consider the specific underlying structure of the control problem. The authors may want to proceed more carefully when utilizing FVI as the benchmark and perform comparison for tasks like pendulum for which model-based LQR-type algorithm can easily excel.

**Questions:**

The experiment provided in the work is only restricted to very low-dimensional control problem, i.e., pendulum. Since this work has exploited the underlying linear-quadratic structure to a great extent, it is more worth looking at the capacity and efficiency of the algorithm on high-dimensional tasks, with both state and action space in large dimensions. Otherwise, the contribution is limited.

---

> ### Author Response · Authors · 2023-11-20
> **Response to Reviewer**
>
> > The authors proposed a new method exclusively for solving LQR problems [...]
>
> > The authors specifically studied an important class of control problem, namely the affine nonlinear LQR problem
>
> > It is questionable if the method in this work can be fairly compared to other general purpose RL methods or ADP methods, since the latter typically won't consider the specific underlying structure of the control problem.
>
> * Thank you for reviewing our paper. We are afraid that the classification of the proposed method as an "affine *nonlinear Linear* Quadratic Regulator problem" is inaccurate, as there is no such thing in either RL or optimal control.
>
> * Please see our General Point 1 above for further discussion of this point.
>
> > The proposed method indeed outperform in tasks where underlying dynamics are known and deterministic.
>
> * Please see our General Point 2 above for further discussion of this point.
>
> > The authors may want to proceed more carefully when utilizing FVI as the benchmark and perform comparison for tasks like pendulum for which model-based LQR-type algorithm can easily excel.
>
> * Respectfully, we struggle to fathom how more carefully we could choose a system to benchmark FVI than to select the **identical** pendulum system chosen by the FVI authors for benchmarking in **both** the original cFVI study (Lutter et al., 2023a) and rFVI study (Lutter et al., 2022).
>
> * We explicitly discuss this point in Table 2 of the manuscript.
>
> * We would like to point out that our work studies swing-up of the pendulum, where the system nonlinearities are at their strongest -- far from chosing a comparison for which our algorithm can "easily excel."
>
> > The experiment provided in the work is only restricted to very low-dimensional control problem, i.e., pendulum.
>
> * Our choice of environments is objectively SOTA in CT-RL upon comparison to the leading deep RL FVI environments in Table 2, including with respect to system order:
>
>   * **SOTA Deep RL FVIs:**
>     * 2nd order: Pendulum
>     * 4th order: Cart Pendulum
>     * 4th order: Furatura Pendulum
>
>   * **RCI:**
>     * 2nd order: Pendulum -- identical to FVIs as benchmark
>     * 4th order: Jet Aircraft (model parameters from NASA wind tunnel data)
>     * 4th order: Ground Robot (model parameters from system ID on actual hardware)
>
> > This work failed to mention other similar works in continuous-time LQR [...]
>
> * Again, we emphasize that RCI is a full nonlinear learning algorithm, not an LQR learning algorithm. Thus, the respective LQR subsections of the two works referenced do not constitute "similar works" to this one.
>
> * However, even the nonlinear control sections do not furnish direct comparability to the proposed method, for the following reasons:
>
>   * Regarding J. Kim, et al. (2021): This algorithm studies a discounted cost formulation, which requires an entirely different theoretical framework to prove convergence, optimality, and closed-loop stability and does not furnish direct numerical comparability to the undiscounted problem studied. Furthermore, the algorithm requires discretizing the system in order to apply its Q-learning framework. Thus, this is not a properly *continous-time* learning algorithm.
>
>   * Regarding H. Wang, et al. (2020): As is stated in the title, this algorithm requires a stochastic control formulation, which also requires an entirely separate theoretical structure and machinery than the deterministic optimal control problem studied. As with J. Kim, et al. (2021), this work also studies a discounted problem. Furthermore, this algorithm also requires applying limiting arguments to a discretized system and control problem. Thus, this is not a properly *continous-time* learning algorithm.

---

> > ### Comment · Reviewer_QT7M · 2023-11-20
> >
> > I appreciate the reply from the authors to my raised concerns and the two general points in the rebuttal. It still seems to me the equation (1) is not a fully nonlinear system; the change of state dynamics depends linearly on control u, even if both f and g can be fully nonlinear. In addition, the objective function (2) is clearly quadratic in u. This system qualifies for the well-studied linear-quadratic (in u) paradigm, for which, as I mentioned, extensive researches and methods have been proposed. In other words, the nonlinearity in x is not as challenging as the nonlinearity in u, thereby, if the authors consider solving a standard LQR problem as in (1)-(2), a more in-depth discussion of various methods may be necessary.

---

> ### Author Response · Authors · 2023-11-21
> **In short, we solving a nonlinear optimal control problem, not "a standard LQR problem as in (1)-(2)". Please check out any optimal control textbook.**
>
> > I appreciate the reply from the authors to my raised concerns and the two general points in the rebuttal. It still seems to me the equation (1) is not a fully nonlinear system; the change of state dynamics depends linearly on control u, even if both f and g can be fully nonlinear.
>
> > In other words, the nonlinearity in x is not as challenging as the nonlinearity in u, thereby, if the authors consider solving a standard LQR problem as in (1)-(2), a more in-depth discussion of various methods may be necessary.
>
> We thank the reviewer for the follow up questions.
>
> * In short, we solving a nonlinear optimal control problem, not "a standard LQR problem as in (1)-(2)".
>
> * In regards to what “nonlinearity” the reviewer considers "challenging" - 1) The reviewer probably did not realize that continuous time RL (CT-RL) cannot be mixed up with discrete-time (DT-RL). CT-RL is truly more challenging than DT-RL. 2) The reviewer probably has not read our manuscript about our clear justifications and discussions that our results are SOTA, and these results are based on SOTA CT-RL problem formulation (1)-(2).
>
> * More specifically, as we state in boldface in the manuscript, and in our General Points 1: We address the **same** affine nonlinear system structure $(f, g)$ as the SOTA CT-RL works in ADP and Deep RL FVIs. This structure is the **standard** in CT-RL.
>
> * Now to shed some new light on the reviewer’s question, given that the SOTA ADP and Deep RL works address the same affine nonlinear system $(f, g)$, should we tell them all that they should be solving LQR problems instead, too?
>
> * As for "a more in-depth discussion of various methods may be necessary" -- We are afraid that we have been thorough, from system under consideration, method, and environments to results, our work is SOTA. If the reviewer has specific references that "require further discussion in the manuscript", please do inform us.
>
> > In addition, the objective function (2) is clearly quadratic in u. This system qualifies for the well-studied linear-quadratic (in u) paradigm, for which, as I mentioned, extensive researches and methods have been proposed.
>
> * We would like to emphasize again that Q-R cost on nonlinear system $\neq$ LQR.
>
> * For the Reviewer's confusion of linear and nonlinear control problems, we respectfully encourage that they revisit a standard optimal control text; e.g., Frank Lewis's "Optimal Control".
>
> * Please see our response to Reviewer 2 regarding Q-R cost. This quadratic cost structure is also the standard in optimal control (linear and nonlinear) dating back to Kalman in the 1960s.
>
> * Q-R cost is also the predominant cost structure chosen by leading ADP CT-RL algorithms; see, e.g., (Vrabie \& Lewis, 2009), (Vamvoudakis \& Lewis, 2010), (Jiang \& Jiang, 2014), (Bian \& Jiang, 2022). The deep RL FVI works (Lutter et al., 2023a), (Lutter et al., 2022) also accommodate Q-R cost.

---

### Author Response · Authors · 2023-11-20
**General Points of Feedback**

We greatly appreciate the AC and Reviewers who provided their valuable time and feedback. We address the following general points here.

## 1). RCI is a Full Nonlinear Learning Algorithm.

Some Reviewers comment that RCI is an LQR algorithm and/or it performs its learning on a linear approximation of the nonlinear system. These claims are, unfortunately, incorrect -- RCI is a full nonlinear learning algorithm. The formulation fully accommodates the system nonlinearities and model uncertainty. We discuss these points in multiple locations throughout the manuscript:

* On the very first sentence of our Method Section 2, we state:

> RCI addresses the **same** affine nonlinear system as the above SOTA ADP and DRL methods:

\begin{align}
    \dot{x} = f(x) + g(x) u
\\hspace{.5in}(1)
   \end{align}

  We even boldfaced "**same**" in the submission. We are unsure how this point could be made more clear to the reader.

* We included explicit language at multiple points along the steps of the algorithm formulation to make clear to the reader that RCI addresses the full nonlinear system (1). E.g., on pp. 3:

> [In Equation (9)], the drift term $\tilde{f}(x) \triangleq f(x) - A x \in \mathbb{R}^{n}$ fully captures 1) the system nonlinearities, and 2) possible model uncertainties [...]

> We emphasize that the equation (9) is still exact to the original nonlinear dynamics (1).

* We even include explicit language in our key convergence, optimality, and closed-loop stability guarantees of Theorem 2.1 that they ``apply to the full nonlinear system (1)."

## 2). RCI Addresses the Same Affine Nonlinear System Structure as the SOTA CT-RL Works in Deep RL and ADP, Requires Less Dynamical Knowledge Than SOTA Deep RL FVIs.

Some Reviewers remark that RCI requires a comprehensive knowledge of the system dynamics, and they claim that the known deterministic structure $(f, g)$ is a weakness of RCI. However:

* Far from a weakness, such an affine nonlinear dynamical structure $(f, g)$ is a standard formulation the RL-based optimal control problem, including the leading deep RL FVI methods (Lutter et al., 2023a, Lutter et al., 2022).

* In fact, our proposed RCI algorithm requires *significantly less* dynamical information of the environment than the leading deep RL FVIs, a point we make objective and explicit in Table 1 of the submission. Definitively:

  * **RCI:** $f, g$
  * **SOTA Deep RL FVIs:** $f, g$, $\partial f/\partial x$, $\partial g/\partial x$, $\partial f/\partial \theta$, $\partial g/\partial \theta$

---

### Meta-Review · Area_Chair_pcu6 · 2023-12-06

**Metareview:**

*Summary*: This paper presents a new continuous-time reinforcement learning (CTRL) algorithm for the control of affine nonlinear systems. The key idea is to use reference command input (RCI) as probing noise in learning.

*Strength*: (1) Rigorous formulation with performance guarantees. (2) Careful evaluations and comparisons with other SOTA CTRL algorithms.

*Weakness*: (1) Writing needs improvement. In particular, the problem formulation needs more justification. Why CTRL in the first place? Why choose this specific formulation in CTRL? In particular, more justifications than "we have fewer assumptions than FVI SOTAs" are needed to convince the general audience why this paper focuses on CTRL problems with exactly known affine dynamics. (2) The "SOTA baselines" considered in this paper are fairly narrow in the context of ICLR and the whole MBRL community.

**Justification For Why Not Higher Score:**

See the weakness part.

**Justification For Why Not Lower Score:**

See the strength part.

---

### Decision · Program_Chairs · 2024-01-16

Reject